# WHAT AND HOW DOES IN-CONTEXT LEARNING LEARN? BAYESIAN MODEL AVERAGING, PARAMETERIZATION, AND GENERALIZATION

## ABSTRACT

In this paper, we conduct a comprehensive study of In-Context Learning (ICL) by addressing several open questions: (a) What type of ICL estimator is learned by large language models? (b) What is a proper performance metric for ICL and what is the error rate? (c) How does the transformer architecture enable ICL? To answer these questions, we adopt a Bayesian view and formulate ICL as a problem of predicting the response corresponding to the current covariate, given a number of examples drawn from a latent variable model. To answer (a), we show that, without updating the neural network parameters, ICL implicitly implements the Bayesian model averaging algorithm, which is proven to be approximately parameterized by the attention mechanism. For (b), we analyze the ICL performance from an online learning perspective and establish a $\mathcal{O}(1/T)$ regret bound for perfectly pretrained ICL, where $T$ is the number of examples in the prompt. To answer (c), we show that, in addition to encoding Bayesian model averaging via attention, the transformer architecture also enables a fine-grained statistical analysis of pretraining under realistic assumptions. In particular, we prove that the error of pretrained model is bounded by a sum of an approximation error and a generalization error, where the former decays to zero exponentially as the depth grows, and the latter decays to zero sublinearly with the number of tokens in the pretraining dataset. Our results provide a unified understanding of the transformer and its ICL ability with bounds on ICL regret, approximation, and generalization, which deepens our knowledge of these essential aspects of modern language models.

## 1 INTRODUCTION

With the ever-increasing sizes of model capacity and corpus, Large Language Models (LLM) have achieved tremendous successes across a wide range of tasks, including natural language understanding (Dong et al., 2019; Jiao et al., 2019), symbolic reasoning (Wei et al., 2022c; Kojima et al., 2022), and conversations (Brown et al., 2020; Ouyang et al., 2022). Recent studies have revealed that these LLMs possess immense potential, as their large capacity allows for a series of *emergent abilities* (Wei et al., 2022b; Liu et al., 2023). One such ability is In-Context Learning (ICL), which enables an LLM to learn from just a few examples, without changing the network parameters. That is, after seeing a few examples in the prompt, a pretrained language model seems to comprehend the underlying concept and is able to extrapolate the understanding to new data points.

Despite the tremendous empirical successes, theoretical understanding of ICL remains limited. Specifically, existing works fail to explain why LLMs the ability for ICL, how the attention mechanism is related to the ICL ability, and how pretraining influences ICL. Although the optimality of ICL is investigated in Xie et al. (2021) and Wies et al. (2023), these works both make unrealistic assumptions on the pretrained models, and their results cannot demystify the particular role played by the attention mechanism in ICL.

In this work, we focus on the scenario where a transformer is first pretrained on a large dataset and then prompted to perform ICL. Our goal is to rigorously understand why the practice of "pretraining + prompting" unleashes the power of ICL. To this end, we aim to answer the following three questions: **(a)** What type of ICL estimator is learned by LLMs? **(b)** What are suitable performance

metrics to evaluate ICL accurately and what are the error rates? **(c)** What is the role played by the transformer architecture during the pretraining and prompting stages? The first and the third questions demand scrutinizing the transformer architecture to understand how ICL happens during transformer prompting. The second question then requires statistically analyzing the extracted ICL process. Moreover, the third question necessitates a holistic understanding beyond prompting — we also need to characterize the statistical error of pretraining and how this error affects prompting.

To address these questions, we adopt a Bayesian view and assume that the examples fed into a pre-trained LLM are sampled from a latent variable model parameterized by a hidden concept $z_* \in \mathfrak{Z}$. Moreover, the pretrained dataset contains sequences of examples from the same latent variable model, but with the concept parameter $z \in \mathfrak{Z}$ itself randomly distributed according to a prior distribution. We mathematically formulate ICL as the problem of predicting the response of the response corresponding to the current covariate, where the prompt contains $t$ examples of covariate-response pairs and the current covariate.

Under such a setting, to answer **(a)**, we show that the perfectly pretrained LLMs perform ICL in the form of Bayesian Model Averaging (BMA). That is, LLM first computes a posterior distribution of $z_* \in \mathfrak{Z}$ given the first $t$ examples, and then predicts the response of the $(t+1)$-th covariate by aggregating over the posterior (Proposition 4.1).

In addition, to answer **(b)**, we adopt the online learning framework and define a notion called ICL regret, which is the averaged prediction error of ICL on a sequence of covariate-response examples. We prove that the ICL regret after prompting $t$ examples is $\mathcal{O}(1/t)$ up to the statistical error of the pretrained model (Theorem 6.2).

Finally, to answer **(c)**, we elucidate the role played by the transformer architecture in prompting and pretraining respectively. In particular, we show that a variant of attention mechanism encodes BMA in its architecture, which enables the transformer to perform ICL via prompting. Such an attention mechanism can be viewed as an extension of linear attention and coincides with the standard softmax attention (Garnelo and Czarnecki, 2023) when the length of the prompt goes to infinity. And thus we show that softmax attention Vaswani et al. (2017) approximately encodes BMA (Proposition 4.3). Besides, the transformer architecture enables a fine-grained analysis of the statistical error incurred by pretraining. In particular, applying the PAC-Bayes framework, we prove that the error of the pretrained language model, measured via total variation, is bounded by a sum of approximation error and generalization error (Theorem 5.3). The approximation error decays to zero exponentially fast as the depth of the transformer increases (Proposition 5.4), while the generalization error decays to zero sublinearly with the number of tokens in the pretraining dataset. This features the first pretraining analysis of transformers in total variation distance, which also takes the approximation error into account. Furthermore, as an interesting extension, we also study the misspecified case where the response variables of the examples fed into the LLM are perturbed. We provide sufficient conditions for ICL to be robust to the perturbations and establish the finite-sample statistical error (Proposition H.4).

In sum, by addressing questions **(a)**–**(c)**, we provide a unified understanding of the ICL ability of LLMs and the particular role played by the attention mechanism. Our theory provides a holistic theoretical understanding of the regret, approximation, and generalization errors of ICL.

## 2 RELATED WORK

**In-Context Learning.** After Brown et al. (2020) showcased the in-context learning (ICL) capacity of GPT-3, there has been a notable surge in interest towards enhancing and comprehending this particular ability (Dong et al., 2022). The ICL ability has seen enhancements through the incorporation of extra training stages (Min et al., 2021; Wei et al., 2021; Iyer et al., 2022), carefully selecting and arranging informative demonstrations (Liu et al., 2021; Kim et al., 2022; Rubin et al., 2021; Lu et al., 2021), giving explicit instructions (Honovich et al., 2022; Zhou et al., 2022b; Wang et al., 2022), and prompting a chain of thoughts (Wei et al., 2022c; Zhang et al., 2022b; Zhou et al., 2022a). In efforts to comprehend the mechanisms of ICL ability, researchers have also conducted extensive work. Empirically, Chan et al. (2022) demonstrated that the distributional properties, including the long-tailedness, are important for ICL. Garg et al. (2022) investigated the function class that ICL can approximate. Min et al. (2022) showed that providing wrong mappings between the input-output

pairs in examples does not degrade the ICL. Theoretically, Akyürek et al. (2022), von Oswald et al. (2022), Bai et al. (2023), and Dai et al. (2022) indicated that ICL implicitly implements the gradient descent or least-square algorithms from the function approximation perspective. However, the first three works only showed that transformers are able to approximate these two algorithms, which may not align with the pretrained model. The last work ignored the softmax module, which turns out to be important in practical implementation. Feng et al. (2023) derived the impossibility results of ICL and the advantage of chain-of-thought for the function approximation. Li et al. (2023) viewed ICL from the multi-task learning perspective and derived the generalization bound. Hahn and Goyal (2023) built the linguistic model for sentences and used the description length to bound the ICL error with this model. Xie et al. (2021) analyzed ICL within the Bayesian framework, assuming the access to the nominal language distribution and that the tokens are generated from Hiddn Markov Model (HMM)s. However, the first assumption hides the relationship between pretraining and ICL, and the second assumption is restrictive. Following this thread, Wies et al. (2023) relaxed the HMM assumption and assumed access to a pretrained model that is close to the nominal distribution conditioned on any token sequence, which is also unrealistic. Two recent works Wang et al. (2023), and Jiang (2023) also provide the Bayesian analysis of ICL. Unfortunately, these Bayesian works cannot explain the importance of the attention mechanism for ICL and clarify how pretraining is related to ICL. In contrast, we prove that the attention mechanism enables BMA by encoding it in the network architecture and we relate the pretraining error of transformers to the ICL regret.

## 3 PRELIMINARY

**Notation.** We denote $\{1, \cdots, N\}$ as $[N]$. For a Polish space $\mathcal{S}$, we denote the collection of all the probability measures on it as $\Delta(\mathcal{S})$. The total variation distance between two distributions $P, Q \in \Delta(\mathcal{S})$ is $\text{TV}(P, Q) = \sup_{A \subseteq \mathcal{S}} |P(A) - Q(A)|$. The $i^{\text{th}}$ entry of a vector $x$ is denoted as $x_i$ or $[x]_i$. For a matrix $X \in \mathbb{R}^{T \times d}$, we index its $i^{\text{th}}$ row and column as $X_{i,:}$ and $X_{:,i}$ respectively. The $\ell_{p,q}$ norm of $X$ is defined as $\|X\|_{p,q} = (\sum_{i=1}^{d} \|X_{:,i}\|_p^q)^{1/q}$, and the *Frobenius norm* of it is defined as $\|X\|_{\text{F}} = \|X\|_{2,2}$.

**Attention and Transformers.** Attention mechanism has been the most powerful and popular neural network module in both Computer Vision (CV) and Natural Language Processing (NLP) communities, and it is the backbone of the LLMs (Devlin et al., 2018; Brown et al., 2020). Assume that we have a query vector $q \in \mathbb{R}^{d_k}$. With $T$ key vectors in $K \in \mathbb{R}^{T \times d_k}$ and $T$ value vectors in $V \in \mathbb{R}^{T \times d_v}$, the attention mechanism maps the query vector $q$ to $\text{attn}(q, K, V) = V^\top \text{softmax}(Kq)$, where $\text{softmax}$ normalizes a vector via the exponential function, i.e., for $x \in \mathbb{R}^d$, $[\text{softmax}(x)]_i = \exp(x_i)/\sum_{j=1}^{d} \exp(x_j)$ for $i \in [d]$. The output is a weighted sum of $V$, and the weights reflect the closeness between $W$ and $q$. For $t$ query vectors, we stack them into $Q \in \mathbb{R}^{t \times d_k}$. Attention maps these queries using the function $\text{attn}(Q, K, V) = \text{softmax}(QK^\top)V \in \mathbb{R}^{t \times d_v}$, where $\text{softmax}$ is applied row-wisely. In the practical design of transformers, practitioners usually use Multi-Head Attention (MHA) instead of single attention to express sophisticated functions, which forwards the inputs through $h$ attention modules in parallel and outputs the sum of these sub-modules. Here $h \in \mathbb{N}$ is a hyperparameter. Taking $X \in \mathbb{R}^{T \times d}$ as the input, MHA outputs $\text{mha}(X, W) = \sum_{i=1}^{h} \text{attn}(XW_i^Q, XW_i^K, XW_i^V)$, where $W = (W_i^Q, W_i^K, W_i^V)_{i=1}^{h}$ is the parameters set of $h$ attention modules, $W_i^Q \in \mathbb{R}^{d \times d_h}$, $W_i^K \in \mathbb{R}^{d \times d_h}$, and $W_i^V \in \mathbb{R}^{d \times d}$ for $i \in [h]$ are weight matrices for queries, keys, and values, and $d_h$ is usually set to be $d/h$ (Michel et al., 2019). The transformer is the concatenation of the attention modules and the fully-connected layers, which is widely adopted in LLMs (Devlin et al., 2018; Brown et al., 2020).

**Large Language Models and In-Context Learning.** Many LLMs are *autoregressive*, such as GPT (Brown et al., 2020). It means that the model continuously predicts future tokens based on its own previous values. For example, starting from a token $x_1 \in \mathfrak{X}$, where $\mathfrak{X}$ is the alphabet of tokens, a LLM $\mathbb{P}_\theta$ with parameter $\theta \in \Theta$ continuously predicts the next token according to $x_{t+1} \sim \mathbb{P}_\theta(\cdot \,|\, S_t)$ based on the past $S_t = (x_1, \cdots, x_t)$ for $t \in \mathbb{N}$. Here, each token represents a word and the position of the word (Ke et al., 2020), and the token sequences $S_t$ for $t \in \mathbb{N}$ live in the sequences space $\mathfrak{X}^*$. LLMs are first *pretrained* on a huge body of corpus, making the prediction $x_{t+1} \sim \mathbb{P}_\theta(\cdot \,|\, S_t)$ accurate, and then prompted to perform downstream tasks. During the pretraining phase, we aim to maximize the conditional probability $\mathbb{P}_\theta(x \,|\, S)$ over the nominal next token $x$ (Brown et al., 2020).

After pretraining, LLMs are prompted to perform downstream tasks without tuning parameters. Different from the finetuned models that learn the task explicitly (Liu et al., 2023), LLMs can implicitly

learn from the examples in the *prompt*, which is known as ICL (Brown et al., 2020). Concretely, pretrained LLMs are provided with a prompt $\texttt{prompt}_t = (\widetilde{c}_1, r_1, \ldots, \widetilde{c}_t, r_t, \widetilde{c}_{t+1})$ with $t$ examples and a query as inputs, where each pair $(\widetilde{c}_i, r_i) \in \mathfrak{X}^* \times \mathfrak{X}$ is an example of the task, and $\widetilde{c}_{t+1}$ is the query, as shown in Figure 8 in Appendix E. For example, the $\texttt{prompt}_t$ with $t = 2$ can be "Cats are animals, pineapples are plants, mushrooms are". Here $\widetilde{c}_1 \in \mathfrak{X}^*$ is a token sequence "Cats are", while $r_1$ is the response "animals". The query $\widetilde{c}_{t+1}$ is "mushrooms are", and the desired response is "fungi". The prompts are generated from a hidden concept $z_* \in \mathfrak{Z}$, e.g., $z_*$ can be the classification of biological categories, where $\mathfrak{Z}$ is the concept space. The generation process is $\widetilde{c}_i \sim \mathbb{P}(\cdot \mid \widetilde{c}_1, r_1, \cdots, \widetilde{c}_{i-1}, r_{i-1}, z_*)$ and $r_i \sim \mathbb{P}(\cdot \mid \texttt{prompt}_{i-1}, z_*)$ for the nominal distribution $\mathbb{P}$ and $i \in [t]$. Thus, when performing ICL, LLMs aim to estimate the conditional distribution $\mathbb{P}(r_{t+1} \mid \texttt{prompt}_t, z_*)$. It is widely conjectured and experimentally found that the pretrained LLMs can implicitly identify the hidden concept $z_* \in \mathfrak{Z}$ from the examples, and then perform ICL by outputting from $\mathbb{P}(r_{t+1} \mid \texttt{prompt}_t, z_*)$. In the following, we will provide theoretical justifications for this claim. We note that delimiters are omitted in our work, and our results can be generalized to handle this case. Since LLMs are autoregressive, the definition of the notation $\mathbb{P}(\cdot \mid S)$ with $S \in \mathfrak{X}^*$ may be ambiguous because the length of the subsequent tokens is not specified. Unless explicitly specified, we let $\mathbb{P}(\cdot \mid S)$ denote the distribution of the next single token conditioned on $S$.

## 4 IN-CONTEXT LEARNING VIA BAYESIAN MODEL AVERAGING

In this section, we show that LLMs perform ICL implicitly via BMA. Given a sequence $S = \{(\widetilde{c}_t, r_t)\}_{t=1}^T$ with $T$ examples generated from a hidden concept $z_* \in \mathfrak{Z}$, we use $S_t = \{(\widetilde{c}_i, r_i)\}_{i=1}^t$ to represent the first $t$ ICL examples in the sequence. Here $\widetilde{c}_t$ and $r_t$ respectively denote the ICL covariate and response. During the ICL phase, a LLM is sequentially prompted with $\texttt{prompt}_t = (S_t, \widetilde{c}_{t+1})$ for $t \in [T-1]$, i.e., the first $t$ examples and the $(t+1)$-th covariate. The prompted LLM aims to predict the response $r_{t+1}$ based on $\texttt{prompt}_t = (S_t, \widetilde{c}_{t+1})$ whose true distribution is $r_{t+1} \sim \mathbb{P}(\cdot \mid \texttt{prompt}_t, z_*)$. For the analysis of ICL, we focus on the following latent variable model

$$r_t = f(\widetilde{c}_t, h_t, \xi_t), \quad \forall t \in [T], \tag{4.1}$$

where the hidden variable $h_t \in \mathcal{H}$ determines the relation between $c_t$ and $r_t$, $\xi_t \in \Xi$ for $t \in [T]$ are i.i.d. random noises, and $f : \mathcal{X} \times \mathcal{H} \times \Xi \to \mathfrak{X}$ is a function that relates response $r_t$ to $\widetilde{c}_t, h_t$, and $\xi_t$. In the data generation process, a hidden concept $z_* \in \mathfrak{Z}$ is first generated from $\mathbb{P}(z)$. The hidden variables $\{h_t\}_{t=1}^T$ are then a stochastic process whose distribution is determined by the hidden concept $z_*$, that is

$$\mathbb{P}(h_t = \cdot \mid \widetilde{c}_t, \{r_\ell, h_\ell, \widetilde{c}_\ell\}_{\ell < t}) = g_{z_*}(h_1, \ldots, h_{t-1}, \zeta_t)$$

for some function $g_{z_*}$ parameterized by $z^*$, where $\{\zeta_t\}_{t=1}^T$ are exogenous noises. The response $r_t$ is then generated according to (4.1). The model in (4.1) essentially assumes that the hidden concept $z_*$ implicitly determines the transition of the conditional distribution $\mathbb{P}(r_t = \cdot \mid \widetilde{c}_t)$ by affecting the evolution of the latent variables $\{h_t\}_{t \in [T]}$, and it does not impose any assumption on the distribution of $\widetilde{c}_t$. This model is quite general, and it subsumes the models in previous works. When $f$ is the emission function in HMM and $h_t = h$ for $t \in [T]$ is the values of hidden states that depend on $z$, model in (4.1) recovers the HMM assumption in Xie et al. (2021). When $h_t = z$ for $t \in [T]$ degenerate to the hidden concept, this recovers the casual graph model in Wang et al. (2023) and the ICL model in Jiang (2023).

Assuming that the tokens follow the statistical model given in (4.1), during pretraining, we collect $N_p$ independent trajectories by sampling from (4.1) with concept $z$ randomly sampled from $\mathbb{P}(z)$. Intuitively, during pretraining, by training in an autoregressive manner, the LLM approximates the conditional distribution $\mathbb{P}(r_{t+1} \mid \texttt{prompt}_t) = \mathbb{E}_{z \sim \mathbb{P}(z)}[\mathbb{P}(r_{t+1} \mid \texttt{prompt}_t, z)]$, which is the conditional distribution of $r_{t+1}$ given $\texttt{prompt}_t$, aggregated over the randomness of the concept $z_*$.

Under the model in (4.1), we will show that pretrained LLMs are able to perform ICL because they secretly implement BMA (Wasserman, 2000) during prompting. For ease of presentation, we first consider the setting where the LLM is *perfectly pretrained*, i.e., the conditional distribution induced by the LLM is given by $\mathbb{P}(r_{t+1} \mid \texttt{prompt}_t)$. We relax this condition by analyzing the pretraining error in Section 5.

**Proposition 4.1** (LLMs Perform BMA)**.** Under the model in (4.1), it holds that

$$\mathbb{P}(r_{t+1} = \cdot \mid \texttt{prompt}_t) = \int \mathbb{P}(r_{t+1} = \cdot \mid \widetilde{c}_{t+1}, S_t, z)\mathbb{P}(z \mid S_t)\mathrm{d}z. \tag{4.2}$$

We note that the left-hand side of (4.2) is the prediction of the pretrained LLM given a prompt $\texttt{prompt}_t$. Meanwhile, the right-hand side is exactly the prediction given by the BMA algorithm that infers the posterior belief of the concept $z_*$ based on $S_t$ and predicts $r_{t+1}$ by aggregating the likelihood in (4.1) with respect to the posterior $\mathbb{P}(z_* = \cdot \,|\, S_t)$. Thus, this proposition shows that perfectly pretrained LLMs are able to perform ICL because they **implement BMA during prompting**. As mentioned, Proposition 4.1 is proved under a more general model than the previous works and thus serves as a generalized result of some claims in the previous works. We note that the claim of Proposition 4.1 is independent of the network structure. This partially explains why LSTMs demonstrate ICL ability in Xie et al. (2021). In the next section, we will demonstrate how the attention mechanism helps to implement BMA. The proof of Proposition 4.1 is in Appendix F.2.

Next, we study the performance of ICL from an online learning perspective. Recall that LLMs are continuously prompted with $S_t$ and aim to predict the $(t + 1)$-th covariate $r_{t+1}$ for $t \in [T-1]$. This can be viewed as an online learning problem. For any algorithm that generates a sequence of density estimators $\{\widehat{\mathbb{P}}(r_t)\}_{t=1}^T$ for predicting $\{r_t\}_{t \in [T]}$, we consider the following ICL regret as its performance metric:

$$\texttt{regret}_t = t^{-1} \sup_z \sum_{i=1}^t \log \mathbb{P}(r_i \,|\, \texttt{prompt}_{i-1}, z) - t^{-1} \sum_{i=1}^t \log \widehat{\mathbb{P}}(r_i). \qquad (4.3)$$

This ICL regret measures the performance of the estimator $\widehat{\mathbb{P}}$ compared with the best hidden concept in hindsight. For the perfectly trained LLMs, the estimator is exactly $\widehat{\mathbb{P}}(r_t) = \mathbb{P}(r_{t+1} \,|\, \texttt{prompt}_t)$. By building the equivalence of pretrained LLM and BMA, we have the following corollary, which shows that predicting $\{r_t\}_{t \in [T]}$ by iteratively prompting the LLM incurs a $\mathcal{O}(1/T)$ regret.

**Corollary 4.2** (ICL Regret of Perfectly Pretrained Model). Under the model in (4.1), we have for any $t \in [T]$ that

$$t^{-1} \sum_{i=1}^t \log \mathbb{P}(r_i \,|\, \texttt{prompt}_{i-1}) \geq \sup_{z \in \mathcal{Z}} \Big( t^{-1} \sum_{i=1}^t \log \mathbb{P}(r_i \,|\, \texttt{prompt}_{i-1}, z) + t^{-1} \log \mathbb{P}_{\mathcal{Z}}(z) \Big).$$

Here $\mathbb{P}_{\mathcal{Z}}$ is the prior of the hidden concept $z \in \mathcal{Z}$. When the hidden concept space $\mathfrak{Z}$ is finite and the prior $\mathbb{P}_{\mathcal{Z}}(z)$ is the uniform distribution on $\mathfrak{Z}$, we have that $\texttt{regret}_t \leq \log |\mathfrak{Z}|/t$. When the nominal concept $z_*$ satisfies that $\sup_z \sum_{i=1}^t \mathbb{P}(r_i \,|\, z, \texttt{prompt}_{i-1}) = \sum_{i=1}^t \mathbb{P}(r_i \,|\, z_*, \texttt{prompt}_{i-1})$ for any $t \in [T]$, the regret is bounded as $\texttt{regret}_t \leq \log(1/\mathbb{P}_{\mathcal{Z}}(z_*))/t$.

This theorem states that the ICL regret of the perfectly pretrained model is bounded by $\log(1/\mathbb{P}_{\mathcal{Z}}(z_*))/t$. This is intuitive since the regret is relatively large if the concept $z_*$ rarely appears according to the prior distribution. This corollary shows that, when given sufficiently many examples, predicting $\{r_t\}_{t \in [T]}$ via ICL is almost as good as the oracle method which knows true concept $z_*$ and the likelihood function $\mathbb{P}(r_i \,|\, \texttt{prompt}_{i-1}, z_*)$. The practical relevance of this result is discussed in Appendix C. The proof of Corollary 4.2 is in Appendix F.3. In Section 5, we characterize the deviation between the learned model and the underlying true model. Next, we show how transformers parameterize BMA.

## 4.1 ATTENTION PARAMETERIZES BAYESIAN MODEL AVERAGING

In the following, we explore the role played by the attention mechanism in ICL. To simplify the presentation, we consider the case where the covariate $\widetilde{c}_t \in \mathfrak{X}^*$ is a single token $c_t \in \mathfrak{X}$ in this subsection. During the ICL phase, pretrained LLMs are prompted with $\texttt{prompt}_t = (S_t, c_{t+1})$ and tasked with predicting the $(t + 1)$-th response $r_{t+1}$. The transformers first separately map the covariates $\widetilde{c}_i$ and responses $r_i$ for $i \in [t]$ to the corresponding feature spaces, which are usually realized by the fully connected layers. We denote these two learnable mappings as $k : \mathbb{R}^d \to \mathbb{R}^{d_k}$ and $v : \mathbb{R}^d \to \mathbb{R}^{d_v}$. Their nominal values are denoted as $k_*$ and $v_*$, respectively. The pretraining of the transformer essentially learns the nominal mappings $v_*$ and $k_*$ with sufficiently many data points. After these transformations, the attention module will take $v_i = v_*(r_i)$ and $k_i = k_*(c_i)$ for $i \in [t]$ as the value and key vectors to predict the result for the query $q_{t+1} = k_{t+1} = k_*(c_{t+1})$. To elucidate the role played by attention, we consider a Gaussian linear simplification of (4.1)

$$v_t = z_* \phi(k_t) + \xi_t, \quad \forall t \in [T], \qquad (4.4)$$

where $\phi : \mathbb{R}^{d_k} \to \mathbb{R}^{d_\phi}$ refers to the feature mapping in some Reproducing Kernel Hilbert Space (RKHS), $z_* \in \mathbb{R}^{d_v \times d_\phi}$ corresponds to the hidden concept, and $\xi_t \sim N(0, \sigma^2 I), t \in [T]$ are i.i.d. Gaussian noises with covariance $\sigma^2 I$. Besides, we assume the prior of $z_*$ is $\mathbb{P}(z)$ is a Gaussian distribution $N(0, \lambda I)$. Note that (4.4) can be written as

$$r_t = v_*^{-1}\Big(z_*\phi\big(k_*(c_t)\big) + \xi_t\Big), \tag{4.5}$$

which is a realization of (4.1) with $h_t = z$, $\xi_t = \epsilon_t$, and $f(c, h, \xi) = v_*^{-1}(h\phi(k_*(c)) + \xi)$. In other words, (4.4), or equivalently (4.5), specifies a specialization of (4.1) where in the feature space, the hidden concept $z_*$ represents a transformation between the value $v$ and the key $k$. Here, we simply take this as the transformation by a matrix, which can be easily generalized by building a bijection between concepts $z$ and complex transformations. In the following, to simplify the notation, let $\mathfrak{K} : \mathbb{R}^{d_k} \times \mathbb{R}^{d_k} \to \mathbb{R}$. denote the kernel function of the RKHS induced by $\phi$. The stacks of the values and keys are denoted as $K_t = (k_1, \dots, k_t)^\top \in \mathbb{R}^{t \times d_k}$ and $V_t = (v_1, \dots, v_t)^\top \in \mathbb{R}^{t \times d_v}$, respectively. Consequently, the model in (4.4) implies that

$$\mathbb{P}(v_{t+1} \mid \mathtt{prompt}_t) = \int \mathbb{P}(v_{t+1} \mid z, q_{t+1})\mathbb{P}(z \mid S_t)\mathrm{d}z \propto \exp\Big(-\big\|v_{t+1} - \bar{z}_t\phi(q_{t+1})\big\|_{\Sigma_t^{-1}}^2/2\Big), \tag{4.6}$$

where we denote by $\Sigma_t$ the covariance of $v_{t+1} \sim \mathbb{P}(\cdot \mid S_t, q_{t+1})$, and the mean concept $\bar{z}_t$ is

$$\bar{z}_t = V_t\big(\phi(K_t)\phi(K_t)^\top + \lambda I\big)^{-1}\phi(K_t) = V_t\big(\mathfrak{K}(K_t, K_t) + \lambda I\big)^{-1}\phi(K_t). \tag{4.7}$$

Combining (4.6) and (4.7), we can see that $\bar{z}_t\phi(q_{t+1})$ essentially measures the similarity between the query and keys, which is quite similar to the attention mechanism defined in Section 3. However, here the similarity is normalization according to (4.7), not by softmax. This motivates us to define a new structure of attention and explore the relationship between the newly defined attention and the original one. For any $q \in \mathbb{R}^{d_k}$, $K \in \mathbb{R}^{t \times d_k}$, and $V \in \mathbb{R}^{t \times d_v}$, we define a variant of the attention mechanism as follows,

$$\mathtt{attn}_\dagger(q, K, V) = V^\top\big(\mathfrak{K}(K, K) + \lambda I\big)^{-1}\mathfrak{K}(K, q). \tag{4.8}$$

From (4.6), (4.7), and (4.8), it holds that the response $v_{t+1}$ for $(t + 1)$-th query is distributed as $v_{t+1} \sim N(\mathtt{attn}_\dagger(q_{t+1}, K_t, V_t), \Sigma_t)$. We note that $\mathtt{attn}_\dagger$ **bakes the BMA algorithm** for the Gaussian linear model **in its architecture**, by first estimating $\bar{z}_t$ via (4.7) and deriving the final estimate from the inner product between $\bar{z}_t$ and $q_{t+1}$. Here $\mathtt{attn}_\dagger(\cdot)$ is an instance of the *intention mechanism* studied in Garnelo and Czarnecki (2023) and can be viewed as a generalization of linear attention. Recall that we define the softmax attention (Vaswani et al., 2017) for any $q \in \mathbb{R}^{d_k}$, $K \in \mathbb{R}^{t \times d_k}$, and $V \in \mathbb{R}^{t \times d_v}$ as $\mathtt{attn}(q, K, V) = V^\top\mathtt{softmax}(Kq)$. In the following proposition, we show that the attention in (4.8) coincides with the softmax attention as the sequence length goes to infinity.

**Proposition 4.3.** We assume that the key-value pairs $\{(k_t, v_t)\}_{t=1}^T$ are independent and identically distributed, and we adopt Gaussian RBF kernel $\mathfrak{K}_{\mathtt{RBF}}$. In addition, we assume that $\|k_t\|_2 = \|v_t\| = 1$. Then, it holds for an absolute constant $C > 0$ and any $q \in \mathbb{R}^{d_k}$ with $\|q\| = 1$ that $\lim_{T\to\infty} \mathtt{attn}_\dagger(q, K_T, V_T) = C \cdot \lim_{T\to\infty} \mathtt{attn}(q, K_T, V_T)$.

The proof is in Appendix F.4. Combined with the conditional probability of $v_{t+1}$ in (4.6), this proposition shows that **softmax attention approximately encodes BMA** in long token sequences (Wasserman, 2000), and thus is able to perform ICL when prompted after pretraining.

## 5 Theoretical Analysis of Pretraining

### 5.1 Pretraining Algorithm

In this section, we describe the pretraining setting. We largely follow the transformer structures in Brown et al. (2020). The whole network is a composition of $D$ sub-modules, and each sub-module consists of a MHA and a Feed-Forward (FF) fully connected layer. Here, $D > 0$ is the depth of the network. The whole network takes $X^{(0)} = X \in \mathbb{R}^{L \times d}$ as its input. In the $t$-th layer for $t \in [D]$, it first takes the output $X^{(t-1)}$ of the $(t-1)$-th layer as the input and forwards it through MHA with a residual link and a layer normalization $\Pi_{\mathrm{norm}}(\cdot)$ to output $Y^{(t)}$, which projects each row of the input into the unit $\ell_2$-ball. Here we take $d_h = d$ in MHA, and the generalization of our result to

general cases is trivial. Then the intermediate output $Y^{(t)}$ is forwarded to the FF module. It maps each row of the input $Y^{(t)} \in \mathbb{R}^{L \times d}$ through the same single-hidden layer neural network with $d_F$ neurons, that is $\mathtt{ffn}(Y^{(t)}, A^{(t)}) = \mathtt{ReLU}(Y^{(t)} A_1^{(t)}) A_2^{(t)}$, where $A_1^{(t)} \in \mathbb{R}^{d \times d_F}$, and $A_2^{(t)} \in \mathbb{R}^{d_F \times d}$ are the weight matrices. Combined with a residual link and layer normalization, it outputs the output of layer $t$ as $X^{(t)}$, that is

$$Y^{(t)} = \Pi_{\mathrm{norm}}\big[\mathtt{mha}(X^{(t-1)}, W^{(t)}) + \gamma_1^{(t)} X^{(t-1)}\big], \quad X^{(t)} = \Pi_{\mathrm{norm}}\big[\mathtt{ffn}(Y^{(t)}, A^{(t)}) + \gamma_2^{(t)} Y^{(t)}\big]. \quad (5.1)$$

Here we allocate weights $\gamma_1^{(t)}$ and $\gamma_2^{(t)}$ to residual links only for the convenience of theoretical analysis. In the last layer, the network outputs the probability of the next token via a softmax module, that is $Y^{(D+1)} = \mathtt{softmax}(\mathbb{I}_L^\top X^{(D)} A^{(D+1)}/(L\tau)) \in \mathbb{R}^{d_y}$, where $\mathbb{I}_L \in \mathbb{R}^L$ is the vector with all ones, $A^{(D+1)} \in \mathbb{R}^{d \times d_y}$ is the weight matrix, $\tau \in (0, 1]$ is the fixed temperature parameter, and $d_y$ is the output dimension. The parameters of each layer are denoted as $\theta^{(t)} = (\gamma_1^{(t)}, \gamma_2^{(t)}, W^{(t)}, A^{(t)})$ for $t \in [D]$ and $\theta^{(D+1)} = A^{(D+1)}$, and the parameter of the whole network is the concatenation of these parameters, i.e., $\theta = (\theta^{(1)}, \cdots, \theta^{(D+1)})$. We consider the transformers with bounded parameters. The set of parameters is

$$\Theta = \Big\{\theta \,\big|\, \big\|A^{(D+1),\top}\big\|_{1,2} \le B_A, \max\big\{|\gamma_1^{(t)}|, |\gamma_2^{(t)}|\big\} \le 1, \big\|A_1^{(t)}\big\|_{\mathrm{F}} \le B_{A,1}, \big\|A_2^{(t)}\big\|_{\mathrm{F}} \le B_{A,2},$$

$$\big\|W_i^{Q,(t)}\big\|_{\mathrm{F}} \le B_Q, \big\|W_i^{K,(t)}\big\|_{\mathrm{F}} \le B_K, \big\|W_i^{V,(t)}\big\|_{\mathrm{F}} \le B_V \text{ for all } t \in [D], i \in [h]\Big\},$$

where $B_A$, $B_{A,1}$, $B_{A,2}$, $B_Q$, $B_K$, and $B_V$ are the bounds of parameter. Here we only consider the non-trivial case where these bounds are larger than 1, otherwise, the magnitude of the output in $D^{\mathrm{th}}$ layer decreases exponentially with growing depth. The probability induced by the transformer with parameter $\theta$ is denoted as $\mathbb{P}_\theta$.

The pretraining dataset consists of $N_{\mathrm{p}}$ independent trajectories. For the $n$-th trajectory with $n \in [N_{\mathrm{p}}]$, a hidden concept $z^n \sim \mathbb{P}_{\mathcal{Z}}(z) \in \Delta(\mathfrak{Z})$ is first sampled, which is the hidden variables of the token sequence to generate, e.g., the theme, the sentiment, and the style. Then the tokens are sequentially sampled from the Markov chain induced by $z^n$ as $x_{t+1}^n \sim \mathbb{P}(\cdot \,|\, S_t^n, z^n)$ and $S_{t+1}^n = (S_t^n, x_{t+1}^n)$, where $x_{t+1}^n \in \mathfrak{X}$, and $S_t^n, S_{t+1}^n \in \mathfrak{X}^*$. Here the Markov chain is defined with respect to the state $S_t^n$, which obviously satisfies the Markov property since $S_i^n$ for $i \in [t-1]$ are contained in $S_t^n$. The pretraining dataset is $\mathcal{D}_{N_{\mathrm{p}}, T_{\mathrm{p}}} = \{(S_t^n, x_{t+1}^n)\}_{n,t=1}^{N_{\mathrm{p}}, T_{\mathrm{p}}}$ where the concepts $z^n$ is hidden from the context and thus unobserved. Here each token sequence is divided into $T_{\mathrm{p}}$ pieces $\{(S_t^n, x_{t+1}^n)\}_{t=1}^{T_{\mathrm{p}}}$. We highlight that this pretraining dataset collecting process subsumes those for GPT, and Masked AutoEncoders (MAE) (Radford et al., 2021). For GPT, each trajectory corresponds to a paragraph or an article in the pretraining dataset, and $z^n \sim \mathbb{P}_{\mathcal{Z}}(z)$ is realized by the selection process of these contexts from the Internet. For MAE, we take $T_{\mathrm{p}} = 1$, and $S_1^n$ and $x_2^n$ respectively correspond to the image and the masked token.

To pretrain the transformer, we adopt the cross-entropy as the loss function, which is widely used in the training of BERT and GPT. The corresponding pretraining algorithm is

$$\widehat{\theta} = \underset{\theta \in \Theta}{\arg\min} -\frac{1}{N_{\mathrm{p}} T_{\mathrm{p}}} \sum_{n=1}^{N_{\mathrm{p}}} \sum_{t=1}^{T_{\mathrm{p}}} \log \mathbb{P}_\theta(x_{t+1}^n \,|\, S_t^n). \quad (5.2)$$

We first analyze the population version of (5.2). In the training set, the conditional distribution of $x_{t+1}^n$ conditioned on $S_t^n$ is $\mathbb{P}(x_{t+1}^n \,|\, S_t^n) = \int_{\mathfrak{Z}} \mathbb{P}(x_{t+1}^n \,|\, S_t^n, z) \mathbb{P}_{\mathcal{Z}}(z \,|\, S_t^n) \mathrm{d}z$, where the unobserved hidden concept is weighed via its posterior distribution. Thus, the population risk of (5.2) is $\mathbb{E}_t[\mathbb{E}_{S_t}[\mathrm{KL}(\mathbb{P}(\cdot \,|\, S_t)\|\mathbb{P}_\theta(\cdot \,|\, S_t)) + H(\mathbb{P}(\cdot \,|\, S_t))]]$, where $t \sim \mathtt{Unif}([T_{\mathrm{p}}])$, $H(p) = -\langle p, \log p \rangle$ is the entropy, and $S_t$ is distributed as the pertaining distribution. Thus, we expect that $\mathbb{P}_\theta$ will converge to $\mathbb{P}$. For MAE, the network training adopts $\ell_2$-loss, and we defer the analysis of this case to Appendix G.4.

## 5.2 Performance Guarantee for Pretraining

We first state the assumptions for the pretraining setting.

**Assumption 5.1.** There exists a constant $R > 0$ such that for any $z \in \mathfrak{Z}$ and $S_t \sim \mathbb{P}(\cdot \,|\, z)$, we have $\|S_t^\top\|_{2,\infty} \le R$ almost surely.

This assumption states that the $\ell_2$-norm of the magnitude of each token in the token sequence is upper bounded by $R > 0$. This assumption holds in most machine learning settings. For BERT and GPT, each token consists of word embedding and positional embedding. For MAE, each token consists of a patch of pixels. The $\ell_2$-norm of each token is bounded in these cases.

**Assumption 5.2.** There exists a constant $c_0 > 0$ such that for any $z \in \mathfrak{Z}$, $x \in \mathfrak{X}$ and $S \in \mathfrak{X}^*$, we have $\mathbb{P}(x \,|\, S, z) \geq c_0$.

This assumption states that the conditional probability of $x$ conditioned on $S$ and $z$ is lower bounded. This comes from the ambiguity of language, that is, a sentence can take lots of words as its next word. Similar regularity assumptions are also widely adopted in ICL literature (Xie et al., 2021; Wies et al., 2023). To state our result, we respectively use $\mathbb{E}_{S \sim \mathcal{D}}$ and $\mathbb{P}_{\mathcal{D}}$ to denote the expectation and the distribution of the average distribution of $S_t^n$ in $\mathcal{D}_{N_{\mathrm{p}}, T_{\mathrm{p}}}$, i.e., $\mathbb{E}_{S \sim \mathcal{D}}[f(S)] = \sum_{t=1}^{T_{\mathrm{p}}} \mathbb{E}_{S_t}[f(S_t)]/T_{\mathrm{p}}$ for any function $f : \mathfrak{X}^* \to \mathbb{R}$.

**Theorem 5.3.** Let $\bar{B} = \tau^{-1} R h B_A B_{A,1} B_{A,2} B_Q B_K B_V$ and $\bar{D} = D^2 d(d_F + d_h + d) + d \cdot d_y$. Under Assumptions 5.1 and 5.2, the pretrained model $\mathbb{P}_{\hat{\theta}}$ by the algorithm in (5.2) satisfies

$$
\mathbb{E}_{S \sim \mathcal{D}} \Big[ \mathrm{TV}\big( \mathbb{P}(\cdot \,|\, S), \mathbb{P}_{\hat{\theta}}(\cdot \,|\, S) \big) \Big]
$$
$$
= O\Bigg( \underbrace{\inf_{\theta^* \in \Theta} \sqrt{\mathbb{E}_{S \sim \mathcal{D}} \mathrm{KL}\big( \mathbb{P}(\cdot|S) \| \mathbb{P}_{\theta^*}(\cdot|S) \big)} + \frac{t_{\mathrm{mix}}^{1/4} \log 1/\delta}{(N_{\mathrm{p}} T_{\mathrm{p}})^{1/4}}}_{\text{approximation error}} + \underbrace{\frac{\sqrt{t_{\mathrm{mix}}}}{\sqrt{N_{\mathrm{p}} T_{\mathrm{p}}}} \Big( \bar{D} \log(1 + N_{\mathrm{p}} T_{\mathrm{p}} \bar{B}) + \log \frac{1}{\delta} \Big)}_{\text{generalization error}} \Bigg)
$$

with probability at least $1 - \delta$, where $t_{\mathrm{mix}}$ is the mixing time of the Markov chains induced by $\mathbb{P}$, formally defined in Appendix G.1.

We define the right-hand side of the equation as $\Delta_{\mathrm{pre}}(N_{\mathrm{p}}, T_{\mathrm{p}}, \delta)$. The first and the second terms in the bound are the **approximation error**. It measures the distance between the nominal distribution $\mathbb{P}$ and the distributions induced by transformers with respect to KL divergence. If the nominal model $\mathbb{P}$ can be represented by transformers exactly, i.e., the realizable case, these two terms will vanish. The third term is the **generalization error**, and it does not increase with the growing sequence length $T_{\mathrm{p}}$. This is proved via the PAC-Bayes framework.

This pretraining analysis is missing in most existing theoretical works about ICL. Xie et al. (2021), Wies et al. (2023), and Jiang (2023) all assume access to an arbitrarily precise pretraining model. Although the generalization bound in Li et al. (2023) can be adapted to the pretraining analysis, the risk definition therein can not capture the approximation error in our result. Furthermore, their analysis cannot fit the maximum likelihood algorithm in (5.2). Concretely, their result can only show that the convergence rate of KL divergence is $O((N_{\mathrm{p}} T_{\mathrm{p}})^{-1/2})$ with a realizable function class. Combined with Pinsker's inequality, this gives the convergence rate for total variation as $O((N_{\mathrm{p}} T_{\mathrm{p}})^{-1/4})$ even in the realizable case.

The deep neural networks are shown to be universal approximators for many function classes (Cybenko, 1989; Hornik, 1991; Yarotsky, 2017). Thus, the approximation error in Theorem 5.3 should vanish with the increasing size of the transformer. To achieve this, we slightly change the structure of the transformer by admitting a bias term in feed-forward modules, taking $A_2^{(t)} \in \mathbb{R}^{d_F \times d_F}$, and admitting $d_F$ to vary across layers. This mildly affects the generalization error by replacing $D \cdot d_F$ by the sum of $d_F$ of all the layers in Theorem 5.3. We derive the approximation error bound when the dimension of each word is equal to one, i.e., $\mathfrak{X} \subseteq \mathbb{R}$. Our method can carry over the case $d > 1$.

**Proposition 5.4** (Informal)**.** Under certain smoothness conditions, if $d_F \geq 16 d_y$, $B_{A,1} \geq 16 R d_y$, $B_{A,2} \geq d_F$ $B_A \geq \sqrt{d_y}$, and $B_V \geq \sqrt{d}$, then for some constant $C > 0$, we have

$$
\inf_{\theta^* \in \Theta} \max_{\|S^\top\|_{2,\infty} \leq R} \mathrm{KL}\big( \mathbb{P}(\cdot \,|\, S) \,\|\, \mathbb{P}_{\theta^*}(\cdot \,|\, S) \big) = O\bigg( d_y \exp\Big( -\frac{C \cdot D^{1/4}}{\sqrt{\log B_{A,1}}} \Big) \bigg).
$$

The formal statement and proof are deferred to Appendix G.3. This proposition states that the **approximation error decays exponentially with the increasing depth**. Combined with this result, Theorem 5.3 provides the full description of the pretraining performance.

# 6 ICL REGRET UNDER PRACTICAL SETTINGS

## 6.1 ICL REGRET WITH AN IMPERFECTLY PRETRAINED MODEL

xIn Section 4, we study the ICL regret with a perfect pretrained model. In what follows, we characterize the ICL regret when the pretrained model has an error. Note that the distribution $\mathcal{D}_{\texttt{ICL}}$ of the prompts of ICL tasks can be different from that of pretraining. We impose the following assumption on their relation.

**Assumption 6.1.** We assume that there exists an absolute constant $\kappa > 0$ such that for any ICL prompt, it holds that $\mathbb{P}_{\mathcal{D}_{\texttt{ICL}}}(\texttt{prompt}) \leq \kappa \cdot \mathbb{P}_{\mathcal{D}}(\texttt{prompt})$.

This assumption states that the prompt distribution is covered by the pretraining distribution. Intuitively, the pretrained model cannot precisely inference on the datapoint that is outside the support of the pretraining distribution. For example, if the pretraining data does not contain any mathematical symbols and numbers, it is difficult for the pretrained model to calculate $2 \times 3$ in ICL precisely. We then have the following theorem characterizing the ICL regret of the pretrained model.

**Theorem 6.2** (ICL Regret of Pretrained Model)**.** We assume that the underlying hidden concept $z_*$ maximizes $\sum_{i=1}^{t} \log \mathbb{P}(r_i \,|\, \texttt{prompt}_{i-1}, z)$ for any $t \in [T]$ and there exists an absolute constant $\beta > 0$ such that $\log(1/p_0(z_*)) \leq \beta$. Under Assumptions 5.1, 5.2, and 6.1, we have with probability at least $1 - \delta$ that

$$\mathbb{E}_{\texttt{prompt} \sim \mathcal{D}_{\texttt{ICL}}}\left[T^{-1} \cdot \sum_{t=1}^{T} \log \mathbb{P}(r_t \,|\, z^*, \texttt{prompt}_{t-1}) - T^{-1} \cdot \sum_{t=1}^{T} \log \mathbb{P}_{\widehat{\theta}}(r_t \,|\, \texttt{prompt}_{t-1})\right]$$
$$\leq \mathcal{O}\big(\beta/T + \kappa \cdot b^* \cdot \Delta_{\text{pre}}(N_{\text{p}}, T_{\text{p}}, \delta)\big).$$

Here we denote by $\Delta_{\text{pre}}(N_{\text{p}}, T_{\text{p}}, \delta)$ the pretraining error in Theorem 5.3.

Theorem 6.2 shows that the expected ICL regret for the pretrained model is upper bounded by the sum of two terms: **(a) the ICL regret for the underlying true model** and **(b) the pretraining error**. These two terms are separately bounded in Sections 4 and 5.

## 6.2 PROMPTING WITH WRONG INPUT-OUTPUT MAPPINGS

In the real-world implementations of ICL, the provided input-output examples may not conform to the nominal distribution induced by $z_*$, and the outputs in examples can be *perturbed*. We temporarily take concept space $\mathfrak{Z}$ as a finite space, and our results can be generalized with a covering number argument. We denote the prompt considered in Section 4 as $\texttt{prompt}_t = (S_t, \widetilde{c}_{t+1})$, $S_t = (\widetilde{c}_1, r_1, \cdots, \widetilde{c}_t, r_t) \in \mathfrak{X}^*$, and $(\widetilde{c}_{i+1}, r_{i+1}) \sim \mathbb{P}(\cdot \,|\, S_i, z_*)$ for $i \in [t-1]$. Here, each input $\widetilde{c}_i \in \mathfrak{X}^l$ is a $l$-length token sequence, and each output $r_i \in \mathfrak{X}$ is a single token. The perturbed prompt is then denoted as $\texttt{prompt}' = (S_t', \widetilde{c}_{t+1})$, where $S_t' = (\widetilde{c}_1, r_1', \cdots, \widetilde{c}_t, r_t') \in \mathfrak{X}^*$, and $r_i'$ for $i \in [t]$ is the modified output. We denote the perturbed prompt distribution as $\mathbb{P}'$. Then the performance of ICL with wrong input-output mappings can be stated as follows.

**Proposition 6.3** (Informal)**.** Under certain assumptions, including the distinguishability assumption $(\min_{z \neq z^*} \texttt{KL}_{\text{pair}}\big(\mathbb{P}(\cdot \,|\, z^*) \,\|\, \mathbb{P}(\cdot \,|\, z)\big) > 2 \log 1/c_0)$, the pretrained model $\mathbb{P}_{\widehat{\theta}}$ in (5.2) predicts the outputs with the prompt containing wrong mappings as

$$\mathbb{E}_{\texttt{prompt}'}\left[\texttt{KL}\big(\mathbb{P}(\cdot \,|\, \widetilde{c}_{t+1}, z_*) \| \mathbb{P}_{\widehat{\theta}}(\cdot \,|\, S_t', \widetilde{c}_{t+1})\big)\right]$$
$$= \mathcal{O}\left(\Delta_{\text{pre}}(N_{\text{p}}, T_{\text{p}}, \delta) + \exp\left(-\frac{\sqrt{t}}{2(1+l)\log 1/c_0}\left(\min_{z \neq z^*} \texttt{KL}_{\text{pair}}\big(\mathbb{P}(\cdot \,|\, z^*) \,\|\, \mathbb{P}(\cdot \,|\, z)\big) + 2 \log c_0\right)\right)\right)$$

with probability at least $1 - \delta$.

The first term is the pretraining error in Theorem 5.3, which is related to the size of the pretraining set and the capacity of the neural networks. The second term is the ICL error. Intuitively, this term represents the concept identification error. If the considered task $z_*$ is distinguishable, i.e., satisfying Assumption H.3, this term decays to $0$ exponentially in $\sqrt{t}$. The required assumptions and formal statement are in Appendix H.2.

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
