# OpenReview forum: "What and How does In-Context Learning Learn? Bayesian Model Averaging, Parameterization, and Generalization"
_ICLR.cc/2024/Conference — Submitted to ICLR 2024_

### Official Review · Reviewer_b1Tv · 2023-10-30

**Soundness:** 3 good
**Presentation:** 2 fair
**Contribution:** 3 good
**Rating:** 5
**Confidence:** 3

**Summary:**

The paper conducts a theoretical investigation into in-context learning (ICL) and addresses three key questions: the choice of ICL estimator, the performance metric for ICL along with its associated error rate, and the role of the transformer architecture in ICL.

**Strengths:**

1.	The paper provides a more general theoretical model on the response and hidden variables compared with previous works in this direction.
2.	The authors propose a new regret metric to quantify the performance of ICL and develops a $\mathcal{O}(1/T)$ regret rate for this metric.
3.	Besides, the authors analyze the error incurred during the pretraining phase, and the error decays exponentially in terms of the depth of the network.

**Weaknesses:**

1.	As this paper primarily focuses on theoretical aspects, it is encouraged to enhance the clarity of the presented proofs. Given that most of the proofs are intense, providing a big picture to roughly explain and outline how the proof works is crucial for readers' comprehension. Otherwise, readers can easily get confused. For instance, in the proof of Theorem 5.3, it would be beneficial to further explain why we should pick the distributions $P$ and $Q$ in this way when we bound the terms in error decomposition. Similarly, in the proof of Proposition 5.4, why should we first construct an approximator for $g^*$, and then $\rho^*$, $\phi^*$? These will be clearer if a big picture is provided.

2.	Following the above point, the authors should explain the reasons from one step to the next step more concretely. For instance, in the proof of Proposition 5.4, the assumptions 5.2 and F.3 do not appear. If there are some steps obtained with these assumptions implicitly, it is encouraged to point it out explicitly. There may be more areas throughout the paper where such improvements in clarity are needed, and the authors are encouraged to address them to enhance overall presentation.

3.	With all these bounds obtained, it is recommended to validate them empirically, even if through synthetic experiments. Such empirical verification would not only enhance the credibility of the theoretical results but also provide an additional perspective on their soundness.

**Questions:**

1.	According to the definition of the assumed model on $r_t$, is it not related to the true hidden concept? Is this a common assumption on realistic data?
2.	Is there any reason that the authors consider the regret in the form of average instead of the usual accumulative regret? Under this new regret, the rate is $\mathcal{O}(1/T)$. Does that mean in terms of the usual accumulative regret, the rate is constant? How do the authors interpret this?
3.	In the proof of Proposition 4.1, how do the authors obtain the first and the third equalities formally? For the current first equality, does it mean $h_{t+1}$ is independent of $c_{t+1}$? Even though the authors mention these equalities come from model (4.2) (by the way, should it be model (4.1) instead?), more concrete explanations are expected.
4.	Regarding the proof of Corollary 4.2, In the first formula, why $P(S_t|z)$ only relates to responses $r_{i}$? In the third formula, should the most outer sum be $\sum_{t=0}^{T}$, or $\sum_{t=1}^{T}$? If the second equality is derived through telescoping, it is not immediately clear why only one term remains. Also, how is the last equality obtained?
5.	Regarding the proof of Theorem 5.3, in the last inequality of (F.2), do we have an expectation over $S_t^n$ for the first term? According to lemma I.5, there is an expectation over $X\sim\nu$ for the squared TV. Also, the authors note the bound of $P_{\hat{\theta}}(x|S) $ in (F.6), but how is this obtained? How is the TV bound located between (F.6) and (F.7) satisfied? Why $ 2\epsilon/b_y$ is in $\mathcal{O}(\frac{1}{NT})$?
6.	In the proof of Proposition 5.4, the authors obtain the bound for the difference in L1 norm to be in terms of $B_{A,1}^{D''}$, which is an exponential growth term, but how does this later become exponentially decay in terms of D? What does it mean for the sentence “We take that D’=… in Lemma I.9”? Do we let D’ and D’’ be these quantities or C in D’ and D’’ be these quantities? How is the TV bound in step 3 satisfied? Please clean up the proof to make it clearer.

---

> ### Author Response · Authors · 2023-11-21
> **Response to Reviewer b1Tv: Part 1**
>
> We thank the reviewer for the careful reading and the valuable feedback. We address the major concerns in the following.
>
> **Added Experimental Results**
>
> We conduct five experiments to verify our theoretical findings, including the Bayesian view (Proposition 4.1 and Eqn. (4.7)), the regret upper-bounded in Corollary 4.2 and  Theorem 6.2, and the constant ratio between $\mathtt{attn}_{\dagger}$ and $\mathtt{attn}$ in Proposition 4.3. The results and implementation details are provided in Appendix C and D in the revised manuscript. We will list a part of numerical results as table below.
>
> * **Verification of the Bayesian View**
>
> Table 1:
>
> |                 | zs-hc | ICL-0 | ICL-1 | ICL-5 | ICL-10 |
> |-----------------|-------|-------|-------|-------|--------|
> | antonym         | 0.736 | 0.000 | 0.296 | 0.651 | 0.675  |
> | country-capital | 0.825 | 0.048 | 0.881 | 0.905 | 0.952  |
> | present-past    | 0.847 | 0.082 | 0.820 | 0.967 | 0.967  |
>
> To verify the Bayesian view that we adopt in the paper, we implemented two experiments. In the first experiment, we explicitly construct the hidden concept vectors that are found by LLMs. Motivated by (4.7) and Proposition 4.3, we construct the hidden concept vector as the average sum over prompts of the values of twenty selected attention heads, i.e., we compress the hidden concept into a vector with dimension $4096$. To demonstrate the effectiveness of the constructed hidden concepts, we add these hidden concept vectors at a layer of LLMs when the model resolves the prompt with zero-shot. In Table 1, "zs-hc" refers to the results of LLMs that infers with learned hidden concept vectors and zero-shot prompt, and "ICL-$i$" refers to the results of LLMs prompted with $i$ examples. This table report the accuracy of LLMs under different prompts. We consider the tasks of finding antonyms, finding the capitals of countries, and finding the past tense of words. The results indicate that the LLMs with learned hidden concept vectors have **comparable performance** with the LLMs prompted with several examples. This indicates that the learned hidden concept vectors are indeed efficient **compression of the hidden concepts**, which proves that LLMs deduce hidden concepts for ICL. This result strongly corroborates with (4.7).
>
> Table 2:
>
> |          | 2 examples | 6 examples | 10 examples | 14 examples | 18 examples |
> |----------|------------|------------|-------------|-------------|-------------|
> | accuracy | 0.000      | 0.917      | 0.917       | 1.000       | 1.000       |
>
>
> In the second experiment, we aim to verify that LLMs implement inference with the Bayesian framework, not with gradient descent [1,2,3] on some tasks. We prompt the LLMs with the history data of a set of similar multi-armed bandit instances with $100$ arms, and let LLMs indicate which arm to pull in a similar new bandit instance. In these similar bandit instances, there is an informative arm, whose reward is exactly the index of the arm with the highest rewards. We also provide the side information that "Some arm may directly tell you the arm with the highest reward, even itself does not have the highest reward". In each example provided in the prompt, there are the rewards of six arms, including the informative arm and the best arm, in one bandit instance. As shown in Table 2, the LLMs can efficiently implement ICL even with only $6$ examples. We note that the gradient descent algorithms in the previous works cannot explain this performance, since the gradient descent algorithms need **at least $100$ data points**, where each data point is the reward of one arm, to learn. In contrast, the Bayesian view can clearly explain Table 2, where LLMs make use of the side information to calculate **a better posterior** for the bandit instance. Thus, LLMs have good ICL performance in this setting.
>
>
> * **Verification of the Regret Bound**
>
> Table 3:
> |                        | 0 examples | 10 examples | 20 examples | 30 examples | 40 examples |
> |------------------------|------------|-------------|-------------|-------------|-------------|
> | $t\cot\text{regret}$   | 1.006      | 8.204       | 10.505      | 10.529      | 10.535      |
> | squared error of query | 1.006      | 0.528       | 0.0217      | 0.001       | 0.001       |
>
> To verify Corollary 4.2 and Theorem 6.2, we implement experiments to evaluate the regret in two settings. In the first setting, the LLMs is trained for the linear regression task from scratch, which is a representative setting studied in [1,4]. The examples in the prompt are $\lbrace(x _i,y _i)\rbrace _{i=1}^{N}$, where $x _i\in\mathbb{R}^d$, $d=20$ and $y _{i}=w^{T}x _{i}$ for some $w$ sampled from Gaussian distribution. Given the Gaussian model, we adopt the squared error to approximate the logarithm of the probability. Then the $t\times \text{regret}$ of the LLMs can be well approximated by the sum of the squared error till time $t$.

---

> > ### Author Response · Authors · 2023-11-21
> > **Response to Reviewer b1Tv: Part 2**
> >
> > The results in Table 3 strongly corroborate our theoretical findings. First, the results verify our claim in Corollary 4.2 and Theorem 6.2 that $t\cdot \text{regret}$ can be **upper bounded by a constant.** Second, the line of squared error indicates that the ICL of LLMs only has a significant error when **$T\leq d$**, i.e., the regret **only increases in this region**. Thus, the regret of the ICL by LLMs is at most linear in $O(d/T)$. From the view of our theoretical result, discretizing the set $\lbrace z\in\mathbb{R}^{d} | \|z\| _{2}\leq d \rbrace$ with approximation error $\delta>0$ will result in a set with $(C/\delta)^{d}$ elements, where $C>0$ is an absolute constant. Corollary 4.2 and Theorem 6.2 imply that the regret is the sum of the **$\log |\mathfrak{Z}|/T = d\log (C/\delta)/T$** and the pretraining error, which matches the simulation results.
> >
> > Table 4:
> > |                    | 50 tokens | 100 tokens | 150 tokens | 200 tokens | 250 tokens |
> > |--------------------|-----------|------------|------------|------------|------------|
> > | $y= x$             | 0.001     | 0.002      | 0.002      | 0.017      | 0.020      |
> > | $y=\sin (x)$       | 0.105     | 3.842      | 3.919      | 4.114      | 4.190      |
> > | $y=x\cdot\sin (x)$ | 0.012     | 0.018      | 0.030      | 0.041      | 0.048      |
> >
> > In the second experiment, we directly evaluate the regret of pretrained LLMs on the function value prediction task. The prompt consists of the values of a function on the points with fixed intervals. Since the values are real numbers, we adopt the method in [5] to transfer a real number to a token sequence. For the pretrained model, we cannot calculate $\mathbb{P}(r _{i}|\text{prompt} _{i-1},z)$ due to the unknown nominal distributions. Thus, we calculate the cumulative negative log-likelihood $\text{CNLL} _{t}=-\sum _{i=1}^{t}\hat{\mathbb{P}}(r _{i}|\text{prompt} _{i-1})$, and $\text{CNLL} _{t}/t$ is an upper bound of the regret. In Table 4, we indicate the cumulative negative log-likelihoods of predicting the values of three functions. The results show that the cumulative negative log-likelihoods are stepped, which means that the cumulative negative log-likelihoods are **upper-bounded by constants in a long period**. This corroborates with Corollary 4.2 and Theorem 6.2.
> >
> > * **Verification of the Constant Ratio between $\mathtt{att}_{\dagger}$ and $\mathtt{att}$**
> >
> > Table 5:
> > |                         | 10 examples | 1000 examples | 2000 examples | 3000 examples | 4000 examples |
> > |-------------------------|-------------|---------------|---------------|---------------|---------------|
> > | 75 th percentile for $d=2$ | 2.312       | 2.347         | 2.327         | 2.299         | 2.297         |
> > | 25 th percentile for $d=2$ | 0.792       | 2.110         | 2.156         | 2.168         | 2.191         |
> > | 75 th percentile for $d=3$ | 2.535       | 3.406         | 3.388         | 3.355         | 3.330         |
> > | 25 th percentile for $d=3$ | 0.569       | 2.904         | 2.957         | 3.032         | 3.078         |
> >
> > To verify Proposition 4.3, we directly calculate the ratio between $\mathtt{att} _{\dagger}$ and $\mathtt{att}$. We consider the case $d _v = 1 $ and $d _k=d$ for some $d>0$. The entries in $K$ of (4.8) are i.i.d. samples of Gaussian distribution, and the $i-$th entry of $V$ is calculated as the inner product between a Gaussian vector and the $i-$th column. Table 5 shows the results for $d=2$ and $d=3$. It shows that the ratio between $\mathtt{att} _{\dagger}$ and $\mathtt{att}$ will converge to a constant. This constant depends on the dimension $d$, which originates from Proposition F.1.

---

> ### Author Response · Authors · 2023-11-21
> **Response to Reviewer b1Tv: Part 3**
>
> **Explanations about the Distribution Construction in the Proof of Theorem 5.3 and the Network Construction in the Proof of Proposition 5.4**
>
> We thank the reviewer for the suggestions. We have modified correspondingly in the revised version, and we added two figures in the revised manuscript to illustrate these two constructions. Here we provide the explanations about the details in the proofs of Theorem 5.3 and Proposition 5.4.
>
> * For Theorem 5.3, we adopt the Pac-Bayes framework to derive the generalization error bound, i.e., Lemma I.2. We restate the main equation in the lemma
>
> \begin{align*}
>         \bigg|\mathbb{E} _{Q}\bigg[\mathbb{E} _{X}\big[ f(X) \big]-\frac{1}{n}\sum _{i=1} ^{n}f(X _{i})\bigg]\bigg|\leq \lambda c\mathbb{E} _{Q}\big[\mu(f)\big]+\frac{1}{n\lambda}\bigg[\text{KL}(Q\|P _{0})+\log\frac{2}{\delta}\bigg].
> \end{align*}
>
> To derive the uniform generalization bound for the transformer function class, we need to set $Q$ and $P_0$ such that the left-hand side of the inequality is close to the generalization error of any function and the right-hand side remains bounded. To achieve this goal, we set **$Q$ as the uniform distribution on the neighborhood of any function**, and the difference between the left-hand side of the inequality and the generalization error of any function can be upper bounded via Proposition F.1. We set $P_0$ as the **uniform distribution on the whole function class**, which guarantees that the right-hand side of the inequality can be upper bounded by $O(\log (NT))$.
>
> * For Proposition 5.4, we bound the approximation error with the help of Theorem 2 in [6], which states that for any permutation-invariant function $g^{\ast}(X):\mathfrak{X}^{L}\rightarrow \mathbb{R}^{d _{y}}$, we have that
> \begin{align*}
>     g^{\ast}(X)=\rho^{\ast}\bigg(\frac{1}{L}\sum _{i=1}^{L}\phi^{\ast}(x _{i})\bigg),
> \end{align*}
> where $\rho^{\ast}: \mathbb{R}\rightarrow \mathbb{R}^{d _{y}}$ and $\phi^{\ast}:\mathfrak{X}\rightarrow \mathbb{R}$. Thus, to approximate a permutation-invariant function, we only **need to approximate $\phi^\ast$ and each component $\rho_i^*$ of $\rho^\ast$**. In the proof, we first state that if we already have $\phi^\ast$ and $\rho^\ast$ approximated by submodules of transformer, then we can approximate $\rho^{\ast}\bigg(\frac{1}{L}\sum _{i=1}^{L}\phi^{\ast}(x _{i})\bigg)$ by **combining these submodules with a single attention** (Step 1). In the second step, we indicate how to **approximate  $\phi^\ast$ and $\rho^\ast$** with the modules in transformers (Step 2).
>
> **Where Assumptions 5.2 and F.3 are used**
>
> We point to where Assumptions 5.2 and F.3 are used in the revised manuscript. In fact, we construct $\phi^\ast$ and $\rho^\ast$ with the help of Lemma I.9, which requires Assumption F.3. In Step 3, we bound the ratio between $\mathbb{P}(x|S)$ and $\mathbb{P} _{\theta^{\ast}}(x|S)$, which requires the denominator to be larger than $0$. We use Assumption 5.2 here to derive the bound.
>
> **Explanations of (4.1)**
>
> We highlight that our model in (4.1) indicates that $r _t$ **depends on the true hidden concept $z _\ast$** through $h _t$, but $c _t$ does not depend on $z _\ast$. In fact, $r _t$ directly depends on some hidden variable $h_t$, whose distribution is determined by $z _\ast$. When $h _{t}=z$ for $t\in[T]$ degenerates to the hidden concept, this recovers the casual graph model in [7] and the model in [8].
>
> **Motivation for the Regret Definition and Why $t\cdot \text{regret}$ is Upperbounded by a Constant**
>
> We define the regret in the form of an average to indicate the **average regret of each example** in the prompt. In the classical supervised learning task, it is common to take mean square error as the performance metric. In the Gaussian model, our regret degenerates to the mean square error. It is also reasonable to consider the cumulative regret, which can be upper bounded by a constant. It originates from the fact that LLMs can identify the correct hidden concept with the error **exponentially decaying in the number of samples $t$**, which is shown in Proposition 6.3. The intuition can be seen from the following approximate calculations under a simpler model. Now we assume that $\lbrace(r _t,c _t)\rbrace _{t=1}^{T}$ are i.i.d. samples generated from $P(r|c,z _\ast)\cdot P(c)=P(r,c| z _\ast)$ . Then we have
>
> \begin{align*}
> 	\frac{1}{T}\sum _{t=1}^{T}\log\frac{P(r _t, c _t|z _\ast)}{P(r _t, c _t|z)}\rightarrow \text{KL}(P(\cdot|z _\ast)\|P(\cdot|z)).
> \end{align*}
> It implies that
> \begin{align*}
> 	\frac{P(z| S _T)}{P(z _\ast| S _T)}=\frac{P(z)P(S _T|z)}{P(z _\ast)P(S _T|z _\ast)}\rightarrow \frac{P(z)}{P(z _\ast)}\exp(-T\cdot\text{KL}(P(\cdot|z _\ast)\|P(\cdot|z))).
> \end{align*}
>
>
> Since $\sum _{t=1}^{T}\exp(-kt)<O(1)$ for $k>0$ can be upper bounded by a constant, LLMs achieve the constant accumulative regret. It is also verified by our **simulation results** in the revised manuscript.

---

> ### Author Response · Authors · 2023-11-21
> **Response to Reviewer b1Tv: Part 4**
>
> **Explanations of The Proof of Proposition 4.1**
>
> We thank the reviewer for the suggestions and modified the revised version accordingly. We provide detailed calculations of the first and the third equality here.
>
> * For the first equality, we have
> \begin{align*}
> \mathbb{P}(r _{t+1}| \text{pt} _t) &= \int \mathbb{P}(r _{t+1}| h _{t+1},\text{pt} _t) \mathbb{P}(h _{t+1}| \text{pt} _t)\mathrm{d}h _{t+1} \\
> &= \int \mathbb{P}(r _{t+1}| h _{t+1},\tilde{c} _{t+1}) \mathbb{P}(h _{t+1}\,|\, S _t)\mathrm{d}h _{t+1},
> \end{align*}
> where the first equality results from the Bayes’ rule, and the second equality results from the fact that $r _{t+1}$ is **conditionally independent** with the previous history given $h _{t+1},\tilde{c} _{t+1}$ and the fact that $h _{t+1}$ only **parameterizes the transition kernel** of $r _{t+1}$ given $c _{t+1}$. Thus, $h _{t+1}$ and $c _{t+1}$ are independent.
>
> * For the third equality,
> \begin{align*}
> &\int \mathbb{P}(r _{t+1} | c _{t+1}, h _{t+1}) \mathbb{P}(h _{t+1}  | S _t, z)\mathbb{P}(z  | S _t) \mathrm{d} h _{t+1} \mathrm{d} z \\
> &\quad= \int \mathbb{P}(r _{t+1} | c _{t+1}, h _{t+1},S _t, z) \mathbb{P}(h _{t+1}  | S _t, z) \mathrm{d} h _{t+1} \mathbb{P}(z  | S _t) \mathrm{d} z \\
> &\quad= \int\mathbb{P}(r _{t+1}|c _{t+1}, S _t, z)\mathbb{P}(z | S _t) \mathrm{d} z,
> \end{align*}
> where the first equality results from the fact that $r _{t+1}$ is conditionally independent with the other variables given $h _{t+1},\tilde{c} _{t+1}$, and the second equality results from the Bayes’ rule.
>
> **Explanations of the Proof of Corollary 4.2**
>
> *In the first formula, since the hidden variable $z$ only parameterizes the **conditional probability** of $r _t$ given $c _t$, $c _t$ and $z$ are independent.
>
> * In the third formula, the most outer sum is $\sum _{t=0}^{T}$. For the telescoping argument, we follow the convention that $\sum _{t=1}^{0}\cdot = 0$. In fact, when $t=0$,
> \begin{align*}
> \mathbb{P}(r _{t+1}|c _{t+1},S _{t})= \frac{\int \mathbb{P}(r _1|c _1,z)\mathbb{P}(z)\mathrm{d}z}{1}.
> \end{align*}
>
>
> Since $\log 1 =0$, **this term just disappears**. Thus, the telescoping argument only leaves one term.
>
>
> * The last equality follows from the direct calculations. Here we present the calculation process for the finite $\mathfrak{Z}$. For any function $f:\mathfrak{Z}\rightarrow \mathbb{R}$ and distribution $p\in\Delta(\mathfrak{Z})$, we will calculate
> \begin{align*}
> 	\min _{q\in\Delta(\mathfrak{Z})} -\sum _{z}f(z)q(z)+\sum _{z}q(z)\log\frac{q(z)}{p(z)}.
> \end{align*}
> Define the lagrangian function
> \begin{align*}
> 	g(q,\lambda) = -\sum _{z}f(z)q(z)+\sum _{z}q(z)\log\frac{q(z)}{p(z)}+\lambda (\sum _{z}q(z)-1).
> \end{align*}
> By solving the equations
> \begin{align*}
> 	\nabla _{q} g(q,\lambda)=0,\quad \frac{\partial}{\partial \lambda}g(q,\lambda)=0,
> \end{align*}
> we have that the minimizer is
> \begin{align*}
> 	q^{\ast}(z) = \frac{p(z)\exp(f(z))}{\sum _{z^\prime}p(z^\prime)\exp(f(z^\prime))}.
> \end{align*}
> Plugging this into the objective function, we have
> \begin{align*}
> 	\min _{q\in\Delta(\mathfrak{Z})} -\sum _{z}f(z)q(z)+\sum _{z}q(z)\log\frac{q(z)}{p(z)} = -\log \sum _{z}p(z)\exp(f(x)).
> \end{align*}
> This justifies the last equality of our proof.
>
> **Explanations of The Details in the Proof of Theorem 5.3**
>
> * In the last inequality of (F.2), we note that there should be no expectation for the first term. The reason is that the left-hand side of (F.1) takes conditional expectation with respect to $\tilde{\mathcal{D}}$ conditioned on $\mathcal{D}$. Since **$S _t^n$ is a part of $\mathcal{D}$**, taking the conditional expectation with respect to $\tilde{\mathcal{D}}$ conditioned on $\mathcal{D}$ of the first term is trivial. Thus, there is no explicit expectation over the first term.
>
>
> * The bound in (F.6) originates from the last **softmax layer** in transformers, which is defined below (5.1). Here $B_A$ is the bound of each component of the softmax input. To derive the lower bound of the output probability of transformers, we calculate the minimal output component of softmax given the bound of each component.
>
>
> * The TV bound located between (F.6) and (F.7) originates from Proposition F.1 and the distribution assignment (F.5). As discussed in the revised manuscript, we set the distribution $P$ as the uniform distribution on the neighborhood on a parameter with **radius proportional to $1/NT$**. The corresponding approximation error measured in $l_1$ distance,i.e., the TV distribution, is bounded by $1/NT$. Thus, Proposition F.1 shows that the approximation error is $O(1/NT)$ a.s. We also added a figure in the appendix of the revised manuscript to highlight this.

---

> ### Author Response · Authors · 2023-11-21
> **Response to Reviewer b1Tv: Part 5**
>
> **Explanations of Details in the proof of Proposition 5.4**
>
> We would like to provide detailed calculations for the exponential decaying error on $D$, and we have modified the revised manuscript accordingly. The main idea is that: (1) the later modules will **amplify** the approximation error in the previous modules (2) we will **balance the depths** of different modules to handle the amplification.
>
>
> * For the problems regarding Step 2:
>
>
>  Lemma I.9 indicates the approximation error $\varepsilon$ of a fully-connected network will depth $D$ can be upper bounded as
> \begin{align*}
> 	\varepsilon\leq \exp\bigg(-\sqrt{\frac{D-\log B}{B}}\bigg).
> \end{align*}
> We denote the depth of $\Psi _{\phi^\ast}$  and $\Psi _{\rho^\ast}$ as $D^\prime$ and $D^{\prime\prime}$, respectively. Then the total approximation error is
> \begin{align*}
> 	\text{approx err}&\leq d _y \exp\bigg(-\sqrt{\frac{D^{\prime\prime}-\log B}{B}}\bigg) +d _y B _{A,1}^{D^{\prime\prime}}\exp\bigg(-\sqrt{\frac{D^{\prime}-\log B}{B}}\bigg)\\
> \end{align*}
> where the inequality follows from the triangle inequality in our manuscript,  $B _{A,1}^{D^{\prime\prime}}$ comes from the fact that the later modules will amplify the error in the previous modules in transformers, and $B _{A,1}$ is the lipschitz constant for each layer.
>
> Fact: For any $l>0,c>0$, we have $\exp(-l\sqrt{x-c})=O(\exp(-l\sqrt{x}))$.
>
> Proof: Basic calculations shows that $\exp(-l\sqrt{x-c})\leq m\exp(-l\sqrt{x})$ for any $m>3$ is equivalent to
> \begin{align*}
> 	\frac{l\cdot c}{\sqrt{x}+\sqrt{x-c}}\leq \log m.
> \end{align*}
> This trivially holds when $x$ is large.
>
> Thus, we have that
> \begin{align*}
>     \text{approx err}=O\bigg( d _y \exp\bigg(-\sqrt{\frac{D^{\prime\prime}}{B}}\bigg) +d _y \exp\bigg(\frac{1}{\sqrt{B}}\big[D^{\prime\prime}\sqrt{B}\log B _{A,1} -\sqrt{D^{\prime}}\big]\bigg)\bigg).
> \end{align*}
> To handle the second term in the right-hand side of this inequality, we require that
> \begin{align*}
>     k\cdot D^{\prime\prime}-\sqrt{D^{\prime}}\leq -\sqrt{D^{\prime\prime}},
> \end{align*}
> where $k= \sqrt{B}\log B _{A,1}$. This is equivalent to
> \begin{align*}
>     D^{\prime}\geq (k\cdot D^{\prime\prime}+\sqrt{D^{\prime\prime}})^2.
> \end{align*}
> Since $D^{\prime}+D^{\prime\prime}\leq D$, where $D$ is the depth of the whole network, we can set
> \begin{align*}
>     D^{\prime\prime}=\sqrt{D}/(2\sqrt{B}\log B _{A,1}),\quad D^{\prime}=D-1-D^{\prime\prime}\geq D/2+D^{3/4}
> \end{align*}
> when $D$ is large ($C=4$ in our original manuscript). This assignments ensure that $D^{\prime}\geq (k\cdot D^{\prime\prime}+\sqrt{D^{\prime\prime}})^2$. Thus, we have that
> \begin{align*}
>     \text{approx err}=O\bigg( d _y \exp\bigg(-\sqrt{\frac{D^{\prime\prime}}{B}}\bigg)\bigg)=O\bigg(d _{y}\exp\bigg(-\frac{D^{1/4}}{\sqrt{C^{2}B^{2}\log B _ {A,1}}}\bigg)\bigg)
> \end{align*}
> for some constant $C>0$. Here we relax the dependency on $B$ a little for the notational clearness, and the relaxation results from the fact that $B\geq 1$ usually.
>
>
> * For the problem regarding Step 3:
>
>
> We note that the TV bound is exactly the $l_1$-norm, whose upper bound is derived in Step 2.
> ___
> We thank the reviewer again for reading our response. We appreciate it if the reviewer could consider raising the score, if our rebuttal has addressed your concerns.
>
> **Reference**
>
> [1] Akyürek E, Schuurmans D, Andreas J, et al. What learning algorithm is in-context learning? investigations with linear models[J]. arXiv preprint arXiv:2211.15661, 2022.
>
> [2] Von Oswald J, Niklasson E, Randazzo E, et al. Transformers learn in-context by gradient descent[C]//International Conference on Machine Learning. PMLR, 2023: 35151-35174.
>
> [3] Bai Y, Chen F, Wang H, et al. Transformers as Statisticians: Provable In-Context Learning with In-Context Algorithm Selection[J]. arXiv preprint arXiv:2306.04637, 2023.
>
> [4] Garg S, Tsipras D, Liang P S, et al. What can transformers learn in-context? a case study of simple function classes[J]. Advances in Neural Information Processing Systems, 2022, 35: 30583-30598.
>
> [5] Gruver N, Finzi M, Qiu S, et al. Large language models are zero-shot time series forecasters[J]. arXiv preprint arXiv:2310.07820, 2023.
>
> [6] Zaheer M, Kottur S, Ravanbakhsh S, et al. Deep sets[J]. Advances in neural information processing systems, 2017, 30.
>
> [7] Wang X, Zhu W, Wang W Y. Large language models are implicitly topic models: Explaining and finding good demonstrations for in-context learning[J]. arXiv preprint arXiv:2301.11916, 2023.
>
> [8] Jiang H. A latent space theory for emergent abilities in large language models[J]. arXiv preprint arXiv:2304.09960, 2023.

---

> > ### Author Response · Authors · 2023-11-22
> >
> > We hope this message finds you well. As the rebuttal period will close soon, we would like to reach out to you and inquire if you have any additional concerns regarding our paper. We would be immensely grateful if you could kindly review our responses to your comments. This would allow us to address any further questions or concerns you may have before the rebuttal period concludes.

---

### Official Review · Reviewer_MVTm · 2023-11-01

**Soundness:** 3 good
**Presentation:** 2 fair
**Contribution:** 2 fair
**Rating:** 5
**Confidence:** 2

**Summary:**

The paper provides a theoretical analysis of in-context learning (ICL) in transformers used in large language models (LLMs), framing the attention mechanism as a Bayesian Model Averaging (BMA) process. It proposes that LLMs use attention weights to perform BMA, contributing to our understanding of model generalization from few-shot examples. The authors develop a series of propositions and theorems that articulate the role of the attention mechanism in facilitating ICL. Specifically, they demonstrate how the attention weights can be seen as performing a type of BMA. Furthermore, the paper extends the theoretical framework to provide regret bounds for ICL, which quantify how well the learning model performs in comparison to the best possible model chosen in hindsight. It also offers an in-depth error analysis that establishes bounds on the error rates of pretrained models. This work aims to deepen the theoretical foundations of ICL, offering a framework that connects with practical machine-learning challenges.

-----

Post rebuttal: Raising to 5 in light of authors' detailed evaluations. It would be good to see those results incorporated in the next version of the paper.

**Strengths:**

By adopting a Bayesian perspective and formulating ICL as a prediction problem based on examples from a latent variable model, the paper unifies the understanding of transformer architecture's ability to enable ICL.

Propositions and theorems are well-formulated, providing substantial theoretical contributions to the field, such as establishing regret bounds for ICL and demonstrating the parameterization of Bayesian Model Averaging through attention mechanisms.

It includes a fine-grained analysis of pretraining under realistic assumptions, offering a bound on the error of pretrained models.

**Weaknesses:**

The paper lacks empirical validation for its theoretical claims. This is a significant gap, as the practical impact of the theoretical insights on real-world tasks remains unclear without empirical evidence.

Some of the assumptions made for the theoretical analysis are not justified with respect to their applicability in practical scenarios. The paper does not sufficiently discuss how these assumptions can be relaxed without losing the theoretical results.

The complexity of the mathematical content could limit the paper's accessibility to a broader audience. Additionally, there is a lack of clear guidance on the reproducibility of the theoretical results, which is critical for the advancement of the field.

While the paper attempts to address important questions, its insights as compared to the existing works seem very limited at this point.

**Questions:**

- How can the theoretical findings be empirically validated in the absence of practical experiments within the paper?

- Is there a possibility to simplify the presentation to make the paper more accessible to non-experts?

- Could the authors clarify the process of deriving the posterior distributions within the context of ICL and how they relate to the attention weights?

- Can the authors provide more details on the assumptions made for the error analysis and how these assumptions can be justified from a practical standpoint?

- Are there specific mathematical models or theories that could challenge or complement the authors' approach to understanding in-context learning?

---

> ### Author Response · Authors · 2023-11-21
> **Response to Reviewer MVTm: Part 1**
>
> We thank the reviewer for the valuable feedback. We address the major concerns in the following.
>
> **Added Experimental Results**
>
> We conduct five experiments to verify our theoretical findings, including the Bayesian view (Proposition 4.1 and Eqn. (4.7)), the regret upper-bounded in Corollary 4.2 and  Theorem 6.2, and the constant ratio between $\mathtt{attn}_{\dagger}$ and $\mathtt{attn}$ in Proposition 4.3. The results and implementation details are provided in Appendix C and D in the revised manuscript. We will list a part of numerical results as the table below.
>
> * **Verification of the Bayesian View**
>
> Table 1:
>
> |                 | zs-hc | ICL-0 | ICL-1 | ICL-5 | ICL-10 |
> |-----------------|-------|-------|-------|-------|--------|
> | antonym         | 0.736 | 0.000 | 0.296 | 0.651 | 0.675  |
> | country-capital | 0.825 | 0.048 | 0.881 | 0.905 | 0.952  |
> | present-past    | 0.847 | 0.082 | 0.820 | 0.967 | 0.967  |
>
> To verify the Bayesian view that we adopt in the paper, we implemented two experiments. In the first experiment, we explicitly construct the hidden concept vectors that are found by LLMs. Motivated by (4.7) and Proposition 4.3, we construct the hidden concept vector as the average sum over prompts of the values of twenty selected attention heads, i.e., we compress the hidden concept into a vector with dimension $4096$. To demonstrate the effectiveness of the constructed hidden concepts, we add these hidden concept vectors at a layer of LLMs when the model resolves the prompt with zero-shot. In Table 1, "zs-hc" refers to the results of LLMs that infers with learned hidden concept vectors and zero-shot prompt, and "ICL-$i$" refers to the results of LLMs prompted with $i$ examples. This table report the accuracy of LLMs under different prompts. We consider the tasks of finding antonyms, finding the capitals of countries, and finding the past tense of words. The results indicate that the LLMs with learned hidden concept vectors have **comparable performance** with the LLMs prompted with several examples. This indicates that the learned hidden concept vectors are indeed efficient **compression of the hidden concepts**, which proves that LLMs deduce hidden concepts for ICL. This result strongly corroborates with (4.7).
>
> Table 2:
>
> |          | 2 examples | 6 examples | 10 examples | 14 examples | 18 examples |
> |----------|------------|------------|-------------|-------------|-------------|
> | accuracy | 0.000      | 0.917      | 0.917       | 1.000       | 1.000       |
>
>
> In the second experiment, we aim to verify that LLMs implement inference with the Bayesian framework, not with gradient descent [1,2,3] on some tasks. We prompt the LLMs with the history data of a set of similar multi-armed bandit instances with $100$ arms, and let LLMs indicate which arm to pull in a similar new bandit instance. In these similar bandit instances, there is an informative arm, whose reward is exactly the index of the arm with the highest rewards. We also provide the side information that "Some arm may directly tell you the arm with the highest reward, even itself does not have the highest reward". In each example provided in the prompt, there are the rewards of six arms, including the informative arm and the best arm, in one bandit instance. As shown in Table 2, the LLMs can efficiently implement ICL even with only $6$ examples. We note that the gradient descent algorithms in the previous works cannot explain this performance, since the gradient descent algorithms need **at least $100$ data points**, where each data point is the reward of one arm, to learn. In contrast, the Bayesian view can clearly explain Table 2, where LLMs make use of the side information to calculate **a better posterior** for the bandit instance. Thus, LLMs have good ICL performance in this setting.
>
>
> * **Verification of the Regret Bound**
>
> Table 3:
> |                        | 0 examples | 10 examples | 20 examples | 30 examples | 40 examples |
> |------------------------|------------|-------------|-------------|-------------|-------------|
> | $t\cot\text{regret}$   | 1.006      | 8.204       | 10.505      | 10.529      | 10.535      |
> | squared error of query | 1.006      | 0.528       | 0.0217      | 0.001       | 0.001       |
>
> To verify Corollary 4.2 and Theorem 6.2, we implement experiments to evaluate the regret in two settings. In the first setting, the LLMs is trained for the linear regression task from scratch, which is a representative setting studied in [1,4]. The examples in the prompt are $\lbrace(x _i,y _i)\rbrace _{i=1}^{N}$, where $x _i\in\mathbb{R}^d$, $d=20$ and $y _{i}=w^{T}x _{i}$ for some $w$ sampled from Gaussian distribution. Given the Gaussian model, we adopt the squared error to approximate the logarithm of the probability. Then the $t\times \text{regret}$ of the LLMs can be well approximated by the sum of the squared error till time $t$.

---

> ### Author Response · Authors · 2023-11-21
> **Response to Reviewer MVTm: Part 2**
>
> The results in Table 3 strongly corroborate our theoretical findings. First, the results verify our claim in Corollary 4.2 and Theorem 6.2 that $t\cdot \text{regret}$ can be **upper bounded by a constant.** Second, the line of squared error indicates that the ICL of LLMs only has a significant error when **$T\leq d$**, i.e., the regret **only increases in this region**. Thus, the regret of the ICL by LLMs is at most linear in $O(d/T)$. From the view of our theoretical result, discretizing the set $\lbrace z\in\mathbb{R}^{d} | \|z\| _{2}\leq d \rbrace$ with approximation error $\delta>0$ will result in a set with $(C/\delta)^{d}$ elements, where $C>0$ is an absolute constant. Corollary 4.2 and Theorem 6.2 imply that the regret is the sum of the **$\log |\mathfrak{Z}|/T = d\log (C/\delta)/T$** and the pretraining error, which matches the simulation results.
>
> Table 4:
> |                    | 50 tokens | 100 tokens | 150 tokens | 200 tokens | 250 tokens |
> |--------------------|-----------|------------|------------|------------|------------|
> | $y= x$             | 0.001     | 0.002      | 0.002      | 0.017      | 0.020      |
> | $y=\sin (x)$       | 0.105     | 3.842      | 3.919      | 4.114      | 4.190      |
> | $y=x\cdot\sin (x)$ | 0.012     | 0.018      | 0.030      | 0.041      | 0.048      |
>
> In the second experiment, we directly evaluate the regret of pretrained LLMs on the function value prediction task. The prompt consists of the values of a function on the points with fixed intervals. Since the values are real numbers, we adopt the method in [5] to transfer a real number to a token sequence. For the pretrained model, we cannot calculate $\mathbb{P}(r _{i}|\text{prompt} _{i-1},z)$ due to the unknown nominal distributions. Thus, we calculate the cumulative negative log-likelihood $\text{CNLL} _{t}=-\sum _{i=1}^{t}\hat{\mathbb{P}}(r _{i}|\text{prompt} _{i-1})$, and $\text{CNLL} _{t}/t$ is an upper bound of the regret. In Table 4, we indicate the cumulative negative log-likelihoods of predicting the values of three functions. The results show that the cumulative negative log-likelihoods are stepped, which means that the cumulative negative log-likelihoods are **upper-bounded by constants in a long period**. This corroborates with Corollary 4.2 and Theorem 6.2.
>
> * **Verification of the Constant Ratio between $\mathtt{att}_{\dagger}$ and $\mathtt{att}$**
>
> Table 5:
> |                         | 10 examples | 1000 examples | 2000 examples | 3000 examples | 4000 examples |
> |-------------------------|-------------|---------------|---------------|---------------|---------------|
> | 75 th percentile for $d=2$ | 2.312       | 2.347         | 2.327         | 2.299         | 2.297         |
> | 25 th percentile for $d=2$ | 0.792       | 2.110         | 2.156         | 2.168         | 2.191         |
> | 75 th percentile for $d=3$ | 2.535       | 3.406         | 3.388         | 3.355         | 3.330         |
> | 25 th percentile for $d=3$ | 0.569       | 2.904         | 2.957         | 3.032         | 3.078         |
>
> To verify Proposition 4.3, we directly calculate the ratio between $\mathtt{att} _{\dagger}$ and $\mathtt{att}$. We consider the case $d _v = 1 $ and $d _k=d$ for some $d>0$. The entries in $K$ of (4.8) are i.i.d. samples of Gaussian distribution, and the $i-$th entry of $V$ is calculated as the inner product between a Gaussian vector and the $i-$th column. Table 5 shows the results for $d=2$ and $d=3$. It shows that the ratio between $\mathtt{att} _{\dagger}$ and $\mathtt{att}$ will converge to a constant. This constant depends on the dimension $d$, which originates from Proposition F.1.
>
> **Summary of the mathematical content and the Reproducibility of the Results**
>
> We would like to highlight that our main contributions are the **theoretical analysis** of ICL. Our theoretical results mainly have three parts:
>
> * The first part of our results explains how the **perfectly pertained LLMs** implement the ICL via Bayesian model averaging (Proposition 4.1) and how the attention structure enables BMA (Proposition 4.3), where the regret of ICL is also analyzed based on Proposition 4.1 (Corollary 4.2). These results are all validated by our experimental results.
> * The second part of our results is the analysis of the **pretraining error** of the transformers, which is decomposed as the sum of the generalization error and the approximation error(Theorem 5.3). The convergence rate of the approximation error is also provided ( Proposition 5).
> * The last part is the ICL performance analysis of the **pre-trained LLMs**. We provide the ICL regret analysis of LLMs with correct input-output prompts (Theorem 6.2), which is verified by our experiments in Appendix C. In addition, we derive the analysis of LLMs when it is prompted with wrong input-output examples( Proposition 6).

---

> > ### Author Response · Authors · 2023-11-21
> > **Response to Reviewer MVTm: Part 3**
> >
> > We provide the proof of all these theoretical results, which are **easily reproducible**. Our theoretical results are a complement to the large body of literature on the empirical results. In addition, we provide the **simulation results** in the revised manuscript to verify our theoretical findings.
> >
> >
> > **Comparison with Existing Works**
> >
> > We note that there is a large body of empirical work explaining the ICL mechanism of LLMs [6,7,8]. Compared with these works, our paper provides the **theoretical justifications** for some claims in them.
> >
> > * In [6], the authors said that “we find that a small number of attention heads transport a compact representation of the demonstrated task”, which is well supported by Proposition 4.3.
> >
> > * In [7], the authors claimed that “randomly replacing labels in the demonstrations barely hurts performance”, which is supported by Proposition 6.3 under the distinguiability assumption.
> >
> > * In [8], the authors claimed that “TabPFN is a Prior-Data Fitted Network (PFN) and is trained offline once, to approximate Bayesian inference on synthetic datasets drawn from our prior.”, which is instantiation of our Bayesian view in Proposition 4.1.
> >
> >
> > We note that theoretical papers mainly study ICL from two perspectives.
> >
> > * The first one adopts the Bayesian view, where LLMs infer the hidden concept for the prediction  [9,10]. The results in [9] are all based on the hidden Markov model assumption. These papers assume that LLMs are **perfectly pretrained**. In contrast, we do not need the strong hidden Markov model assumption, and we analyze the pretraining error of LLMs.
> > * The other kinds of papers focus on the approximation ability of LLMs, where LLMs run optimization algorithms for prediction[1,2,11]. They analyze how transformers can approximate gradient descent. In contrast, our work proposes a totally different mechanism. We note that [11] proposes that LLMs implement natural gradient descent for ICL. Theorem 2 in [12] and Theorem 1 in [13] show that implementing natural gradient descent in the exponential family can be viewed as the Bayesian inference. This view **agrees with Proposition 4.1** in our paper.
> >
> > **Simplification of the Presentation**
> >
> > We simplified the presentation of our revised manuscript in the following perspectives:
> > * We provide detailed proof guidelines of our theoretical results.
> > * We add the figures to illustrate the important steps in our proof.
> > * We provide the empirical simulations to verify our theoretical findings.
> >
> > **The Process of Deriving the Posterior Distributions**
> >
> > From the theoretical perspective, in the Gaussian linear model, we first show that the hidden concept can be estimated via the attention defined in (4.8), which is the **expectation of the posterior distribution**.  Proposition 4.3 shows that the softmax attention and the attention in (4.8) is the same up to a constant when $T$ is large. The weights of the modules before this attention can be viewed as the parameters of the representation. In summary, our theoretical results in Section 4.1 show that the softmax attention module is able to estimate the hidden concept in the Gaussian linear model.
> >
> > From the practical perspective, in our first experiment, we extract the hidden concept vector as the average sum over prompts of the values of twenty selected attention heads. This hidden concept vector is the **mean embedding of the attention heads** with respect to the likelihood, which contains the information of the posterior distributions. The experimental results show that the LLMs with hidden concept vectors in the zero-shot setting have comparable performance with the LLMs prompted with several examples. This verifies that the constructed hidden concept vector indeed contains the essential information about the posterior distribution of the hidden concept. Similar experiments can be found in [6] and [14] for more settings.
> >
> > **Practical Relevance Of Assumptions**
> >
> > We would like to discuss the relevance of our models and assumptions for the real-world applications:
> >
> > * Assumption 5.1:
> >
> > Assumption 5.1 requires that the norm of the token is bounded. It is satisfied in real applications, where the token is a finite-dimensional vector with bounded components. For example, the tokenizers used in GPT-NeoX-20B [15] and Llama2 [16] both satisfy this assumption.
> >
> > * Assumption 5.2:
> >
> > We remark that Assumption 5.2 applies to real data. The assumption comes from the ambiguity of language, that is, a sentence can take lots of words as its next word. Similar regularity assumptions are also widely adopted in ICL literature [9,17].

---

> ### Author Response · Authors · 2023-11-21
> **Response to Reviewer MVTm: Part 4**
>
> * Assumption 6.1:
> This assumption states the coverage of the pretraining data. We note that there will be an **information-theoretic barrier** without this assumption, the ICL learning problem presents an information-theoretic barrier. For example, if the pretraining data does not contain any material about a specific mathematical symbol in the ICL prompt. In this case, it will be extremely difficult for the LLM to derive the correct prediction, since the meaning of this math symbol is unclear to LLM.
>
> * Assumption G.1:
> This assumption states that the input-output pairs are independent when conditioned on the hidden concept $z$. This assumption is relevant in many applications, since the inputs of demonstrations are **independently sampled**, and the output only depends on the input when the concept (task) $z$ is clear.
>
> * Assumption G.2:
> This assumption states the coverage of the hidden concept $z _\ast$. If $P(z _\ast)=0$, this concept never appears in the pretraining dataset. It will be very difficult for LLMs to complete this task as explained in the previous response. Compared with Assumption 6.1, this assumption is stronger since the coverage is in the hidden space, and it will be used to handle the wrong outputs.
>
> * Assumption G.3:
> This assumption states the distinguishability of the current concept $z _\ast$. When there are only wrong outputs in the prompt, the concept $z _\ast$ itself should be distinguishable enough to **identify from the wrong input-outputs pairs**. This assumption is also stated in [9,17].
>
> ___
> We thank the reviewer again for reading our response. We appreciate it if the reviewer could consider raising the score, if our rebuttal has addressed your concerns.
>
> **Reference**
>
> [1] Akyürek E, Schuurmans D, Andreas J, et al. What learning algorithm is in-context learning? investigations with linear models[J]. arXiv preprint arXiv:2211.15661, 2022.
>
> [2] Von Oswald J, Niklasson E, Randazzo E, et al. Transformers learn in-context by gradient descent[C]//International Conference on Machine Learning. PMLR, 2023: 35151-35174.
>
> [3] Bai Y, Chen F, Wang H, et al. Transformers as Statisticians: Provable In-Context Learning with In-Context Algorithm Selection[J]. arXiv preprint arXiv:2306.04637, 2023.
>
> [4] Garg S, Tsipras D, Liang P S, et al. What can transformers learn in-context? a case study of simple function classes[J]. Advances in Neural Information Processing Systems, 2022, 35: 30583-30598.
>
> [5] Gruver N, Finzi M, Qiu S, et al. Large language models are zero-shot time series forecasters[J]. arXiv preprint arXiv:2310.07820, 2023.
>
> [6] Todd E, Li M L, Sharma A S, et al. Function vectors in large language models[J]. arXiv preprint arXiv:2310.15213, 2023.
>
> [7] Min S, Lyu X, Holtzman A, et al. Rethinking the role of demonstrations: What makes in-context learning work?[J]. arXiv preprint arXiv:2202.12837, 2022.
>
> [8] Hollmann N, Müller S, Eggensperger K, et al. Tabpfn: A transformer that solves small tabular classification problems in a second[J]. arXiv preprint arXiv:2207.01848, 2022.
>
> [9] Xie S M, Raghunathan A, Liang P, et al. An explanation of in-context learning as implicit bayesian inference[J]. arXiv preprint arXiv:2111.02080, 2021.
>
> [10] Jiang H. A latent space theory for emergent abilities in large language models[J]. arXiv preprint arXiv:2304.09960, 2023.
>
> [11]Ahn K, Cheng X, Daneshmand H, et al. Transformers learn to implement preconditioned gradient descent for in-context learning[J]. arXiv preprint arXiv:2306.00297, 2023.
>
> [12] Dai B, He N, Dai H, et al. Provable Bayesian inference via particle mirror descent[C]//Artificial Intelligence and Statistics. PMLR, 2016: 985-994.
>
> [13] Raskutti G, Mukherjee S. The information geometry of mirror descent[J]. IEEE Transactions on Information Theory, 2015, 61(3): 1451-1457.
>
> [14] Hendel R, Geva M, Globerson A. In-context learning creates task vectors[J]. arXiv preprint arXiv:2310.15916, 2023.
>
>
> [15] Black S, Biderman S, Hallahan E, et al. Gpt-neox-20b: An open-source autoregressive language model[J]. arXiv preprint arXiv:2204.06745, 2022.
>
> [16] Touvron H, Martin L, Stone K, et al. Llama 2: Open foundation and fine-tuned chat models[J]. arXiv preprint arXiv:2307.09288, 2023.
>
> [17] N. Wies, Y. Levine, A. Shashua. The learnability of in-context learning. arXiv preprint arXiv:2303.07895 (2023).

---

> > ### Comment · Reviewer_MVTm · 2023-11-21
> > **Thanks**
> >
> > Thanks for the detailed evaluation and results. I have increased my score to 5. I think the discussions and results will significantly strengthen the current paper.

---

> > > ### Author Response · Authors · 2023-11-21
> > >
> > > Thank you for engaging in discussion with us and increasing the scores. We would like to reach out to you and inquire if you have any additional concerns regarding our paper, and we will try our best to address them.

---

### Official Review · Reviewer_xbQq · 2023-11-01

**Soundness:** 4 excellent
**Presentation:** 3 good
**Contribution:** 4 excellent
**Rating:** 6
**Confidence:** 2

**Summary:**

The paper provides a study of in-context learning (ICL) through a Bayesian view and addresses related questions. They first show that large language models perform ICL by adopting a Bayesian model averaging (BMA) algorithm during inference by showing that a prediction for a prompt from a perfectly pretrained model equals the prediction of BMA. Then, to measure the ICL performance, the authors consider ICL regret as a performance metric that measures the difference from prediction to the best hidden concept. The regret is shown to converge with increasingly more examples. To show the connection between the transformer network and the BMA algorithm, the authors define a variant of the attention layers, which then show that the modified attention mechanism encodes the BMA algorithm for the Gaussian linear model. For the sequence length going to infinity, the provided attention variant coincides with the softmax attention.
Additionally, the authors provide the pretraining analysis that can measure the pretraining error with approximation and generalization errors. Under certain smoothness conditions, approximation error converges with bigger models. Finally, for an imperfect pretrained model, the authors can successfully upper-bound the ICL regret.

**Strengths:**

+ The progression and steps make sense.

+ The theoretical results seem to be new and are a nice addition to ICL setting.

+ The theoretical contribution seems to be a generalization for some ICL settings assumed in previous works.

**Weaknesses:**

- How big is the ICL regret in practice? It is hard to tell how tight the bound is in practice. How can it be measured for LLMs?
Since there is no application of the results, it is hard to tell how far the bound is really in practice.

- How can we in practice measure the validation performance of ICL?

**Questions:**

- What is sequentially mean? Are we not really doing sequential but all at once prompting, which is different from iterative prompting?

- What is the "wrong" input-ouput mapping? There is no real wrong or right here. How can you get $z_*$?

- Is KL divergence the best to measure distribution gap, which are key in measuring the generalization gap? What is the embedding space it has to use? Is there a better divergence metrics to measure that?

- Even though the progression and paper is smooth, it would be nice to add visualization to main paper rather than in Appendix.

- The analysis only involves pretraining in relation to the ICL performance. However, large language models also have fine-tuning distribution involved as an "intermediate" between the two aforementioned stages. Should that introduce some +/- gaps in divergence?

- Where do we get the hidden concept in the LLM?

**Minor:**

+ Why LLMs the ability for ICL (page1)

+ Hiddn Markov Model (page 3)

+ Variables are undefined (or defined later than introduced):
  + N and d, d_k,  d_v
  + What is W?
  + Large T and small t difference?
+ Pertaining distribution (page 7)
+ xIn section (page 9)
+ correctly inference (page 9)

---

> ### Author Response · Authors · 2023-11-21
> **Response to Reviewer xbQq: Part 1**
>
> We thank the reviewer for the valuable feedback. We address the major concerns in the following.
>
> **Added Experimental Results**
>
> We conduct five experiments to verify our theoretical findings, including the Bayesian view (Proposition 4.1 and Eqn. (4.7)), the regret upper-bounded in Corollary 4.2 and  Theorem 6.2, and the constant ratio between $\mathtt{attn}_{\dagger}$ and $\mathtt{attn}$ in Proposition 4.3. The results and implementation details are provided in Appendix C and D in the revised manuscript. We will list a part of numerical results as table below.
>
> * **Verification of the Bayesian View**
>
> Table 1:
>
> |                 | zs-hc | ICL-0 | ICL-1 | ICL-5 | ICL-10 |
> |-----------------|-------|-------|-------|-------|--------|
> | antonym         | 0.736 | 0.000 | 0.296 | 0.651 | 0.675  |
> | country-capital | 0.825 | 0.048 | 0.881 | 0.905 | 0.952  |
> | present-past    | 0.847 | 0.082 | 0.820 | 0.967 | 0.967  |
>
> To verify the Bayesian view that we adopt in the paper, we implemented two experiments. In the first experiment, we explicitly construct the hidden concept vectors that are found by LLMs. Motivated by (4.7) and Proposition 4.3, we construct the hidden concept vector as the average sum over prompts of the values of twenty selected attention heads, i.e., we compress the hidden concept into a vector with dimension $4096$. To demonstrate the effectiveness of the constructed hidden concepts, we add these hidden concept vectors at a layer of LLMs when the model resolves the prompt with zero-shot. In Table 1, "zs-hc" refers to the results of LLMs that infers with learned hidden concept vectors and zero-shot prompt, and "ICL-$i$" refers to the results of LLMs prompted with $i$ examples. This table report the accuracy of LLMs under different prompts. We consider the tasks of finding antonyms, finding the capitals of countries, and finding the past tense of words. The results indicate that the LLMs with learned hidden concept vectors have **comparable performance** with the LLMs prompted with several examples. This indicates that the learned hidden concept vectors are indeed efficient **compression of the hidden concepts**, which proves that LLMs deduce hidden concepts for ICL. This result strongly corroborates with (4.7).
>
> Table 2:
>
> |          | 2 examples | 6 examples | 10 examples | 14 examples | 18 examples |
> |----------|------------|------------|-------------|-------------|-------------|
> | accuracy | 0.000      | 0.917      | 0.917       | 1.000       | 1.000       |
>
>
> In the second experiment, we aim to verify that LLMs implement inference with the Bayesian framework, not with gradient descent [1,2,3] on some tasks. We prompt the LLMs with the history data of a set of similar multi-armed bandit instances with $100$ arms, and let LLMs indicate which arm to pull in a similar new bandit instance. In these similar bandit instances, there is an informative arm, whose reward is exactly the index of the arm with the highest rewards. We also provide the side information that "Some arm may directly tell you the arm with the highest reward, even itself does not have the highest reward". In each example provided in the prompt, there are the rewards of six arms, including the informative arm and the best arm, in one bandit instance. As shown in Table 2, the LLMs can efficiently implement ICL even with only $6$ examples. We note that the gradient descent algorithms in the previous works cannot explain this performance, since the gradient descent algorithms need **at least $100$ data points**, where each data point is the reward of one arm, to learn. In contrast, the Bayesian view can clearly explain Table 2, where LLMs make use of the side information to calculate **a better posterior** for the bandit instance. Thus, LLMs have good ICL performance in this setting.
>
>
> * **Verification of the Regret Bound**
>
> Table 3:
> |                        | 0 examples | 10 examples | 20 examples | 30 examples | 40 examples |
> |------------------------|------------|-------------|-------------|-------------|-------------|
> | $t\cot\text{regret}$   | 1.006      | 8.204       | 10.505      | 10.529      | 10.535      |
> | squared error of query | 1.006      | 0.528       | 0.0217      | 0.001       | 0.001       |
>
> To verify Corollary 4.2 and Theorem 6.2, we implement experiments to evaluate the regret in two settings. In the first setting, the LLMs is trained for the linear regression task from scratch, which is a representative setting studied in [1,4]. The examples in the prompt are $\lbrace(x _i,y _i)\rbrace _{i=1}^{N}$, where $x _i\in\mathbb{R}^d$, $d=20$ and $y _{i}=w^{T}x _{i}$ for some $w$ sampled from Gaussian distribution. Given the Gaussian model, we adopt the squared error to approximate the logarithm of the probability. Then the $t\times \text{regret}$ of the LLMs can be well approximated by the sum of the squared error till time $t$.

---

> ### Author Response · Authors · 2023-11-21
> **Response to Reviewer xbQq: Part 2**
>
> The results in Table 3 strongly corroborate our theoretical findings. First, the results verify our claim in Corollary 4.2 and Theorem 6.2 that $t\cdot \text{regret}$ can be **upper bounded by a constant.** Second, the line of squared error indicates that the ICL of LLMs only has a significant error when **$T\leq d$**, i.e., the regret **only increases in this region**. Thus, the regret of the ICL by LLMs is at most linear in $O(d/T)$. From the view of our theoretical result, discretizing the set $\lbrace z\in\mathbb{R}^{d} | \|z\| _{2}\leq d \rbrace$ with approximation error $\delta>0$ will result in a set with $(C/\delta)^{d}$ elements, where $C>0$ is an absolute constant. Corollary 4.2 and Theorem 6.2 imply that the regret is the sum of the **$\log |\mathfrak{Z}|/T = d\log (C/\delta)/T$** and the pretraining error, which matches the simulation results.
>
> Table 4:
> |                    | 50 tokens | 100 tokens | 150 tokens | 200 tokens | 250 tokens |
> |--------------------|-----------|------------|------------|------------|------------|
> | $y= x$             | 0.001     | 0.002      | 0.002      | 0.017      | 0.020      |
> | $y=\sin (x)$       | 0.105     | 3.842      | 3.919      | 4.114      | 4.190      |
> | $y=x\cdot\sin (x)$ | 0.012     | 0.018      | 0.030      | 0.041      | 0.048      |
>
> In the second experiment, we directly evaluate the regret of pretrained LLMs on the function value prediction task. The prompt consists of the values of a function on the points with fixed intervals. Since the values are real numbers, we adopt the method in [5] to transfer a real number to a token sequence. For the pretrained model, we cannot calculate $\mathbb{P}(r _{i}|\text{prompt} _{i-1},z)$ due to the unknown nominal distributions. Thus, we calculate the cumulative negative log-likelihood $\text{CNLL} _{t}=-\sum _{i=1}^{t}\hat{\mathbb{P}}(r _{i}|\text{prompt} _{i-1})$, and $\text{CNLL} _{t}/t$ is an upper bound of the regret. In Table 4, we indicate the cumulative negative log-likelihoods of predicting the values of three functions. The results show that the cumulative negative log-likelihoods are stepped, which means that the cumulative negative log-likelihoods are **upper-bounded by constants in a long period**. This corroborates with Corollary 4.2 and Theorem 6.2.
>
> * **Verification of the Constant Ratio between $\mathtt{att}_{\dagger}$ and $\mathtt{att}$**
>
> Table 5:
> |                         | 10 examples | 1000 examples | 2000 examples | 3000 examples | 4000 examples |
> |-------------------------|-------------|---------------|---------------|---------------|---------------|
> | 75 th percentile for $d=2$ | 2.312       | 2.347         | 2.327         | 2.299         | 2.297         |
> | 25 th percentile for $d=2$ | 0.792       | 2.110         | 2.156         | 2.168         | 2.191         |
> | 75 th percentile for $d=3$ | 2.535       | 3.406         | 3.388         | 3.355         | 3.330         |
> | 25 th percentile for $d=3$ | 0.569       | 2.904         | 2.957         | 3.032         | 3.078         |
>
> To verify Proposition 4.3, we directly calculate the ratio between $\mathtt{att} _{\dagger}$ and $\mathtt{att}$. We consider the case $d _v = 1 $ and $d _k=d$ for some $d>0$. The entries in $K$ of (4.8) are i.i.d. samples of Gaussian distribution, and the $i-$th entry of $V$ is calculated as the inner product between a Gaussian vector and the $i-$th column. Table 5 shows the results for $d=2$ and $d=3$. It shows that the ratio between $\mathtt{att} _{\dagger}$ and $\mathtt{att}$ will converge to a constant. This constant depends on the dimension $d$, which originates from Proposition F.1.
>
> **Validation Performance of ICL**
>
> We are not very clear about the meaning of “validation performance” in the questions. We would like to ask the reviewer to clarify the question. Here we attempt to answer this question with some potential explanations of it.
>
> If “validation performance” means validating the theoretical findings in the empirical results, our simulations in the modified manuscript provide a detailed response to it.  Our simulation results verify the Bayesian view (Proposition 4.1 and (4.7)), Corollary 4.2, and Proposition 4.3. The detailed analysis is provided in Appendix C of the revised manuscript.
>
> If “validation performance” means that we would like to validate the ICL performance of a given LLM, then we can measure the ICL performance of the LLM by splitting the data. Given a set of sentences, we can split them into a training set and a testing set. During the training process, we could implement cross-validation to tune the parameters. The performance of LLMs can be tested on the test set.

---

> ### Author Response · Authors · 2023-11-21
> **Response to Reviewer xbQq: Part 3**
>
> **Explanation of Word "Sequentially"**
>
> The word "sequentially" comes from the fact that we view the ICL from the **online learning perspective**. We would like to understand how the ICL performance scales with an increasing number of examples in the prompt. For example, in Corollary 4.2, we prompt the LLMs with the $\text{prompt}_{t}$ that has $t$ examples in time $t$. We evaluate the regret in this setting. This is the iterative prompting.
>
> **Explanation of Wrong Input-Output Mapping**
>
> The word "wrong" refers to the fact that **input-output pairing is not correct**. We would like to give an example of the wrong input-output pairing. Assume that we want LLMs to predict the classification of biological categories. The prompts with correct input-output mapping can be "Cats are animals, pineapples are plants, mushrooms are", while the prompt with wrong input-output mapping can be "Cats are plants, pineapples are animals, mushrooms are". Empirically, LLMs are able to derive the **correct answer even with random labels**, and the input-output pattern in the prompt is essential for efficient ICL[6]. Here we provide the **theoretical understanding** for these phenomena under the distinguishability assumption. We show that the pretrained LLMs can perform efficient ICL even with wrong labels, including the random label as a special case, when the distinguishability assumption holds.
>
> **Why we use KL divergence**
>
> Since we adopt the cross-entropy loss to train the LLMs, KL divergence is a natural choice for it. When quantifying the training performance, we measure it via the total variation distance of the conditional probability in the **token space**, not in the embedding space. We leave the exploration of other divergence metrics as the future works.
>
> **The Discussion of Finetuning**
>
> We note that the LLMs learned from the pretraining are powerful enough. For example, LLMs are able to efficiently implement the time-series tasks in a zero-shot manner[5]. Then we will discuss the influence of two kinds of finetuning.
>
> The first kind of finetuning is to optimize the parameters of LLMs on a specific task. We note that our performance guarantee in Theorem 5.3 takes expectation with respect to the pretraining distribution. After the fine-tuning process, the distribution here will be closer to the **token distribution of the desired task**. This will improve the performance of LLMs on this task.
>
> The other kind of finetuning is instruction finetuning, which aims to align the behavior of LLMs to the human preference. People first learn a reward function from the human feedback and use the learned reward to tune LLMs with reinforcement learning techniques. The analysis of this process requires the additional techniques about reinforcement learning, and we leave this as future work.
>
> **How to get the hidden concept in the LLMs**
>
> First, we would like to specify the reasons why LLMs can generate the hidden concepts. There are two main reasons: 1. The pretraining data of LLMs comes from the multiple tasks. There are a sufficient number of concepts presented in the pretraining data. 2. The LLMs are large enough to extract the hidden concepts from the prompt. In Proposition 4.3, we theoretically show that the **attention head** contains the hidden concept information in Gaussian linear model.
>
> Second, we would like to discuss how to get the hidden concept in the LLM in experiments. In our experiment section in Appendix C, we extract the **hidden concept vector as the average sum over prompts of the values of twenty selected attention heads**, i.e., we compress the hidden concept into a vector. This is motivated by Proposition 4.3. To demonstrate the effectiveness of the constructed hidden concepts, we add these hidden concept vectors at a layer of LLMs when the model resolves the prompt with zero-shot. The experimental results show that the LLMs with hidden concept vector in the zero-shot setting have comparable performance with the LLMs prompted with several examples. This verifies that the constructed hidden concept vector indeed **contains the essential information about the hidden concept**. This method is recently proposed and tested in [7]. We note that there are other methods to extract the hidden concept vectors. According to [8], we can also learn the hidden concept deduced by LLMs as token values. We can represent the hidden concept as some special tokens. The value of these tokens are trained by maximizing the probability of the correct answer when the LLMs are prompted with some examples. These learned tokens are shown to be effective in the prompt design.
> ___
> We thank the reviewer again for reading our response. We appreciate it if the reviewer could consider raising the score, if our rebuttal has addressed your concerns.

---

> > ### Author Response · Authors · 2023-11-21
> > **Response to Reviewer xbQq: Part 4**
> >
> > **Reference**
> >
> > [1] Akyürek E, Schuurmans D, Andreas J, et al. What learning algorithm is in-context learning? investigations with linear models[J]. arXiv preprint arXiv:2211.15661, 2022.
> >
> > [2] Von Oswald J, Niklasson E, Randazzo E, et al. Transformers learn in-context by gradient descent[C]//International Conference on Machine Learning. PMLR, 2023: 35151-35174.
> >
> > [3] Bai Y, Chen F, Wang H, et al. Transformers as Statisticians: Provable In-Context Learning with In-Context Algorithm Selection[J]. arXiv preprint arXiv:2306.04637, 2023.
> >
> > [4] Garg S, Tsipras D, Liang P S, et al. What can transformers learn in-context? a case study of simple function classes[J]. Advances in Neural Information Processing Systems, 2022, 35: 30583-30598.
> >
> > [5] Gruver N, Finzi M, Qiu S, et al. Large language models are zero-shot time series forecasters[J]. arXiv preprint arXiv:2310.07820, 2023.
> >
> > [6] Min S, Lyu X, Holtzman A, et al. Rethinking the role of demonstrations: What makes in-context learning work?[J]. arXiv preprint arXiv:2202.12837, 2022.
> >
> > [7] Todd E, Li M L, Sharma A S, et al. Function vectors in large language models[J]. arXiv preprint arXiv:2310.15213, 2023.
> >
> > [8] Wang X, Zhu W, Wang W Y. Large language models are implicitly topic models: Explaining and finding good demonstrations for in-context learning[J]. arXiv preprint arXiv:2301.11916, 2023.

---

> ### Author Response · Authors · 2023-11-22
>
> We hope this message finds you well. As the rebuttal period will close soon, we would like to reach out to you and inquire if you have any additional concerns regarding our paper. We would be immensely grateful if you could kindly review our responses to your comments. This would allow us to address any further questions or concerns you may have before the rebuttal period concludes.

---

> > ### Comment · Reviewer_xbQq · 2023-11-22
> >
> > I appreciate the authors' response and additional results.
> >
> > The KL-divergence is used to measure generalization gap. However, the recent work has shown that KL divergence on the token space cannot explain performance across benchmarks [9].
> >
> > How does your result 6.3 relates to findings in [10, 11] that models generally do not perform well on flipped labels and larger models actually follow the input-output mapping rather than "correctly" answering with the ground truth label.
> >
> > For validation performance, I meant test performance, since in real life there is no training or test dataset that the user's query will belong to.
> >
> > [9] Gregory Yauney, Emily Reif, David Mimno, Data Similarity is Not Enough to Explain Language Model Performance, https://arxiv.org/pdf/2311.09006.pdf
> > [10] Katerina Margatina, Timo Schick, Nikolaos Aletras, Jane Dwivedi-Yu, Active Learning Principles for In-Context Learning with Large Language Models, https://arxiv.org/abs/2305.14264
> > [11] Jerry Wei, Jason Wei, Yi Tay, Dustin Tran, Albert Webson, Yifeng Lu, Xinyun Chen, Hanxiao Liu, Da Huang, Denny Zhou, Tengyu Ma, Larger language models do in-context learning differently, https://arxiv.org/abs/2303.03846

---

> > > ### Author Response · Authors · 2023-11-22
> > > **Response to the Questions of Reviewer xbQq: Part 1**
> > >
> > > **Problems about KL Divergence**
> > >
> > > We would like to highlight that the problem studied in [1] is **different** from the problem in our paper. In fact, our pretraining bound in Theorem 5.3 derives the upper bound of $\mathbb{E} _{S\sim\mathcal{D}}[\text{TV}\big(\mathbb{P}( \cdot \,|\,S),\mathbb{P} _{\hat{\theta}}( \cdot \,|\,S)\big)]$. Here $\mathcal{D}$ is the distribution of the pretraining data, which is the mixture of the data distribution from different tasks according to the prior. The total variation distance here and the KL divergence in the right-hand side of the inequality in Theorem 5.3 is adopted to measure the distance between nominal distribution and the LLM. However, the problem studied in [1] is about how the bound of $\mathbb{E} _{S\sim\mathcal{D}}[\text{TV}(\mathbb{P}( \cdot \,|\,S),\mathbb{P} _{\hat{\theta}}( \cdot \,|\,S))]$ could imply for the bound of $\mathbb{E} _{S\sim\tilde{\mathcal{D}}}[\text{TV}(\mathbb{P}( \cdot \,|\,S),\mathbb{P} _{\hat{\theta}}( \cdot \,|\,S))]$, where $\tilde{\mathcal{D}}$ is a data distribution different from $\mathcal{D}$. [1] adopts several ways to measure the **distance between $\tilde{\mathcal{D}}$ and $\mathcal{D}$**. Thus, these are **different two problems**.
> > > ___
> > >
> > > **Explanations of The Experimental Phenomenon**
> > >
> > > We would like to first explain how the size of the network influence the pretraining process and how the pretraining error enter the ICL phase.
> > >
> > > First, in the pretraining phase, the larger model will learn the correct likelihood on more tasks with small prior. Concretely, given sufficient pretraining data, the larger model could have smaller pretraining error, since the approximation error is smaller. This smaller pretraining error will **force the larger LLMs to learn the correct likelihood on task with small prior**. For example, we assume the pretraining errors for a large LLM and a small LLM are $0.001$ and $0.1$, respectively. Given a task with prior $0.01$, the contribution of this task in $\mathbb{E} _{S\sim\mathcal{D}}[\text{TV}(\mathbb{P}( \cdot \,|\,S),\mathbb{P} _{\hat{\theta}}( \cdot \,|\,S))]$ is about $0.01\cdot$ total variation error on this task. Thus, the error of large LLM on this task should be upperbounded by $0.001/0.01=0.1$, while the error of the small LLM only needs to be trivially upperbounded by $0.1/0.01=10$. This means that small LLM can be totally wrong on this task, i.e., from the Bayesian view, the small LLM **cannot correctly identify the task with small prior**.
> > >
> > > Then in the ICL phase, we consider the task in Figure 1 of [2], i.e., return the reverse sentiment. We note that the this task can appear in the corpus as a task in the long tail, i.e., $\kappa$ in Assumption 6.1 can be large. Since the larger LLMs learns the correct likelihood on this task, they will return "negative" for "A smile on your face". In contrast, the smaller LLMs are not guaranteed to perform well. From our bound in Proposition H.4, we need $\kappa\Delta_{\mathrm pre}(N_{\mathrm p},T_{\mathrm p},\delta)$ to be small enough for the accurate ICL, which can only be **achieved by large enough LLMs**.
> > >
> > > We note that the claim "However, the most remarkable finding is the significant performance improvement achieved by selecting similar in-context examples for few-shot learning, particularly in the context of classification tasks." in [3]  supports the Bayesian view in our work. The similar examples can help LLMs to better identify the precise task, i.e., they increase the posterior probaility of the precise task to implement.
> > >
> > > ___
> > >
> > > **Problems About Performance Validation**
> > >
> > >
> > > We would like to highlight the problem raised by the reviewer is a fundamental problem even in the supervised learning setting: we can only guarantee the performance of the learned model **with respect to a distribution**, and there is no guarantee for the performance on **a single point**. In fact, people usually measure the performance of an estiamte with the average error on **a set of data points** which do not appear in the training set.
> > >
> > > For example, we consider the linear regression problem, where $x _i\sim \text{unif}([0,1])$ and $y _i = 2 x _i$ for $i\in[N]$. The goal is to find the function $f$ from a function class $\mathcal{F}$ that is the best approximator of $y=2x$ based on $D=\lbrace (x _i,y _i)\rbrace _{i=1}^{N}$. We consider the function class $\mathcal{F}=\lbrace f_1,f_2\rbrace$ that contains two elements, where $f_1(x) = 2x$ when $x\neq 0.5$, $f_1(0.5)=100$, and $f_2(x)=3x$. Then $P(\hat{f}=f_1)=1$, since $P(x _i = 0.5 \text{ for some }i\in[N])=0$.

---

> ### Author Response · Authors · 2023-11-22
> **Response to the Questions of Reviewer xbQq: Part 2**
>
> To test the performance of the estimate $\hat{f}$, if we sample a single point $x=0.5$, then there is no performance guarantee. However, we can sample a set of points $\tilde{x} _i\sim \text{unif}([0,1])$ for $i\in[\tilde{N}]$. Here $\tilde{x} _i$ for $i\in[\tilde{N}]$ are different from $x _i$ for $i\in[N]$ with probability $1$. Then we have that
> \begin{align*}
>     \frac{1}{\tilde{N}}\sum _{i=1}^{\tilde{N}} (\hat{f}(\tilde{x} _i)-2\tilde{x} _i)^2\rightarrow 0 \text{ in probability as }\tilde{N}\rightarrow\infty.
> \end{align*}
>
> ___
>
> We would like to inquire if you have any additional concerns regarding our paper, and we will try our best to address any further questions or concerns you may have before the rebuttal period concludes.
> ___
>
> **Reference**
>
> [1] Gregory Yauney, Emily Reif, David Mimno, Data Similarity is Not Enough to Explain Language Model Performance
>
>
> [2] Jerry Wei, Jason Wei, Yi Tay, Dustin Tran, Albert Webson, Yifeng Lu, Xinyun Chen, Hanxiao Liu, Da Huang, Denny Zhou, Tengyu Ma, Larger language models do in-context learning differently
>
> [3] Katerina Margatina, Timo Schick, Nikolaos Aletras, Jane Dwivedi-Yu, Active Learning Principles for In-Context Learning with Large Language Models

---

### Meta-Review · Area_Chair_UTGt · 2023-12-14

**Metareview:**

The paper studies in-context learning through a Bayesian lens. They show that in-context learning may be implementing Bayesian model averaging. The rebuttal phase resulted in robust discussion and lots of new experiments to provide additional validation for theoretical results. This will strengthen the paper, but it is my view that it may need to go through a full additional round of reviews.

**Justification For Why Not Higher Score:**

Below borderline.

**Justification For Why Not Lower Score:**

NA

---

### Decision · Program_Chairs · 2024-01-16

Reject